# BI-LEVEL CONTRASTIVE LEARNING FOR KNOWLEDGE ENHANCED MOLECULE REPRESENTATIONS

## ABSTRACT

Molecule representation learning is crucial for various downstream applications, such as understanding and predicting molecular properties and side effects. In this paper, we propose a novel method called GODE, which takes into account the two-level structure of individual molecules. We recognize that molecules have an intrinsic graph structure as well as being a node in a larger molecule knowledge graph. GODE integrates graph representations of individual molecules with multi-domain biochemical data from knowledge graphs. By pre-training two graph neural networks (GNNs) on different graph structures, combined with contrastive learning, GODE fuses molecular structures with their corresponding knowledge graph substructures. This fusion results in a more robust and informative representation, which enhances molecular property prediction by harnessing both chemical and biological information. When fine-tuned across 11 chemical property tasks, our model outperforms existing benchmarks, registering an average ROC-AUC uplift of 12.7% for classification tasks and an average RMSE/MAE enhancement of 34.4% for regression tasks. Impressively, it surpasses the current leading model in molecule property predictions with average advancements of 2.1% in classification and 6.4% in regression tasks.

## 1 INTRODUCTION

Recent years have witnessed a surge of efforts in tailoring machine learning models for chemical and biological data (Wang et al., 2021a; Li et al., 2022; Somnath et al., 2021; Wang et al., 2023). A crucial challenge in this area is coming up with potent representations for molecular structures that are essential for subsequent tasks. (Yang et al., 2019; Haghighatlari et al., 2020). To address this, graph neural networks (GNNs) has been widely deployed to facilitate representation learning (Li et al., 2021; Hu et al., 2019). However, the standard practice of employing molecular graphs as GNN input can unintentionally limit their potential for effective and robust representation.

Molecular data (e.g., chemical and biological datasets), exhibit diverse representational complexities (Tong et al., 2017; Argelaguet et al., 2020). When examining individual molecules, their structures naturally lend themselves to graph representations, wherein atoms become nodes and bonds form the edges. For collections of molecules, their inter-relationships can be encapsulated in knowledge graphs (KGs), with each molecule represented as a unique node. Notable examples of these KGs encompass UMLS (Bodenreider, 2004), PrimeKG (Chandak et al., 2023a), and Pub-ChemRDF (Fu et al., 2015). Stemming from this observation, we put forth the hypothesis that by skillfully integrating these two distinct types of graph data, the individual molecular graphs and the broader KG sub-graphs that center on molecules, we can craft a richer molecule representation. Such an enhanced representation would likely lead to predictions that are more accurate and robust.

Previous attempts have sought to unify molecule structures with knowledge graphs for property prediction. For instance, Ye et al. (2021) combines molecule embeddings with static KG embeddings (Bordes et al., 2013). However, such amalgamations sometimes fail to capture the local information of molecules in the KG, resulting in marginal prediction enhancements. On the other hand, Fang et al. (2022b; 2023) highlight the benefits of improving molecule representations using contrastive learning, supported by their designed chemical element KG. This approach results in more visible performance improvements, showing the value of using KGs with molecular data. Our work aims to find new ways to integrate biochemical knowledge graphs into molecular prediction models.

In this study, we propose a new approach, coined as "**G**raph as a **N**ode" (GODE), designed to pre-train Graph Neural Networks (GNNs). Our approach encompasses bi-level self-supervised tasks, targeting both molecular structures and their corresponding sub-graphs within the knowledge graph (KG). By synergizing this strategy with contrastive learning, (GODE) yields more robust embeddings for molecule property predictions.

Our major contributions can be summarized as follows:

- **A new paradigm to connect knowledge and data**. Our GODE method offers a new paradigm for integrating molecular structures with their corresponding knowledge graphs, which not only yields richer and better molecular representation in our use case but also can be extended to other application domains.

- **Robust embedding enhancement**. For molecular representation, they need to be robust to ensure accurate and consistent predictions across diverse molecular datasets. By integrating information from different domains for the same molecule, our approach leverages the shared knowledge across modalities, ensuring a more comprehensive representation. By employing bi-level self-supervised pre-training with contrastive learning, we significantly enhance the robustness and reliability of the embeddings. Our generated embeddings can provide more accurate predictions on molecular properties, providing a solid foundation for various applications.

- **A new molecular knowledge graph MolKG**. We have constructed MolKG, a comprehensive knowledge graph tailored to molecular data. MolKG encapsulates vast molecular information and facilitates enhanced knowledge-driven molecular analyses.

To evaluate the performance of GODE, we conducted extensive experiments across 11 chemical property prediction tasks. We compared GODE to state-of-the-art methods such as GROVER (Rong et al., 2020), MolCLR (Wang et al., 2021a), and KANO (Fang et al., 2023). Our evaluations demonstrate GODE's superior performance in molecular property prediction, surpassing the baselines by 12.7% and 34.4% for classification and regression tasks, respectively.

## 2 RELATED WORKS

**Graph-based Molecular Representation Learning.**  Over the years, various streams of molecular representation methods have been proposed. They encompass traditional fingerprint-based approaches (Rogers & Hahn, 2010; Jaeger et al., 2018) and modern graph neural network (GNN) methods (Jin et al., 2017; Coley et al., 2019; Jin et al., 2018; Zheng et al., 2019). While Mol2Vec (Jaeger et al., 2018) adopts a molecule interpretation akin to Word2Vec for sentences (Mikolov et al., 2013), it overlooks substructure roles in chemistry. In contrast, GNN-based techniques can overcome this limitation by capturing more insightful details from aggregated sub-graphs. This advantage yields enhanced representations for chemical nodes, bonds, and entire molecules (Cai et al., 2022; Rong et al., 2020; Hu et al., 2019; Wang et al., 2021a). Consequently, our study adopts GNN as the foundational framework for representing molecules.

**Biomedical Knowledge Graphs.**  Various biomedical/biochemical knowledge graphs (KGs) have emerged to capture interconnections among diverse entities like genes, proteins, diseases, and drugs (Belleau et al., 2008; Szklarczyk et al., 2019; Piñero et al., 2020; Fu et al., 2015; Bodenreider, 2004). Notably, PubChemRDF (Fu et al., 2015) spotlights biochemical domains, furnishing machine-readable chemical insights encompassing structures, properties, activities, and bioassays. Its subdivisions (e.g., *Compound*, *Cooccurrence*, *Descriptor*, *Pathway*) amass comprehensive chemical information. PrimeKG (Chandak et al., 2023b) is another KG that provides a multimodal view of precision medicine. Our study has a complementary focus and constructs a molecule-centric KG from those base KGs for supporting molecule property prediction tasks.

**Molecular Property Predictions.** We focus on molecular property prediction, an essential downstream task for chemical representation learning frameworks. Three main aspects of the molecular property attract researchers: quantum mechanics properties (Yang et al., 2019; Liao et al., 2019; Shindo & Matsumoto, 2019; Gilmer et al., 2017), physicochemical properties (Shang et al., 2018; Wang et al., 2019; Bécigneul et al., 2020), and toxicity (Xu et al., 2017; Withnall et al., 2020; Yuan & Ji, 2020; Huang et al., 2020). Most of the recent works on molecular predictions are based on GNN (Duvenaud et al., 2015; Mansimov et al., 2019; Feinberg et al., 2020; 2018). However, the

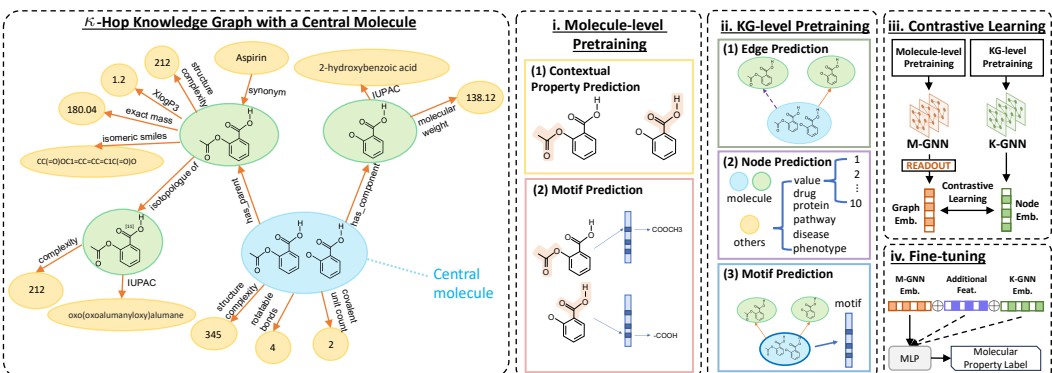

Figure 1: **Overview of our contrastive self-supervised pre-training framework GODE for enhanced molecular representation learning**. *Left*: the $\kappa$-hop KG sub-graph consisting of molecule-relevant relational knowledge, originated by a central molecule. *Right*: We conduct (**i**) **Molecule-level Pre-training** (§3.1) on the molecule graphs with contextual property prediction and motif prediction tasks; (**ii**) **KG-level Pre-training** (§3.2) on the $\kappa$-hop KG sub-graphs of a central molecule with the tasks of edge prediction, node prediction, and motif prediction; (**iii**) **Contrastive Learning** (§3.3) to maximize the agreement between M-GNN and K-GNN, pre-trained by (i) and (ii), respectively; and (**iv**) **Fine-tuning** (§3.4) our learned embedding, optionally enriched with extracted molecular-level features, for specific property predictions.

methods mentioned only focus on chemical structures and do not consider inter-relations among chemicals and knowledge graphs, which could improve property prediction.

**Contrastive Learning in Molecular Representation.** The surge in cross-modality contrastive learning (Radford et al., 2021; Wang et al., 2022b; Yang et al., 2022) has spurred its integration into molecular representation. Noteworthy studies such as (Stärk et al., 2022; Zhu et al., 2022) have harnessed contrastive learning to fuse 3D and 2D molecule representations. This technique has found applications in diverse domains, spanning chemical reactions (Lee et al., 2021; Seidl et al., 2022), natural language (Su et al., 2022; Zeng et al., 2022; Edwards et al., 2021; Seidl et al., 2023), microscopy images (Sanchez-Fernandez et al., 2022), and chemical element knowledge (Fang et al., 2023). Uniquely, our work leverages contrastive learning to facilitate knowledge transfer between biochemical KGs and molecules.

**Fusing Knowledge Graph and Molecules.** Regarding the amalgamation of KG and molecules, Ye et al. (2021) introduced an approach that blends the static KG embedding of drugs with their structural representations for downstream tasks. However, this method overlooks contextual cues around molecule nodes, thus yielding limited performance enhancements. In a different vein, Wang et al. (2022a) proposed a Graph-of-Graph technique, augmenting graph representation to enrich molecular graph information potentially. Yet, strategies such as pre-training and contrastive learning for aligning the same entity across diverse graph modalities remain unexplored. In contrast, Fang et al. (2022b; 2023) pioneered a contrastive learning-based approach, augmenting molecule structures with element-wise knowledge to create an innovative graph structure. This avenue yielded notable advances in molecule property predictions. Unlike existing methods, GODE extracts a molecule's sub-graph from the our molecule-centric KG, offering a new representation that links molecular data and KGs. We provide more details for comparing GODE to its similar works in Appendix F.

## 3 GODE FRAMEWORK

In this section, we present GODE framework. First, we define a few key concepts below.

**Definition 1 (Molecule Graph)** *A molecule graph (MG) is a structured representation of a molecule, where atoms (or nodes) are connected by bonds (or edges). An MG $G_m$ can be viewed as a graph structure with a set of nodes $\mathcal{V}_m$ representing atoms and a set of edges $\mathcal{E}_m$ representing bonds such that $G_m = (\mathcal{V}_m, \mathcal{E}_m)$.*

**Definition 2 (Knowledge Graph)** *A knowledge graph (KG) is a structured representation of knowledge, where entities (or nodes) are connected by relations (or edges). A directed KG can formally be represented as a set of $n$ triples: $\mathcal{T} = \{\langle h, r, t \rangle_i\}_i^n$ where each triple contains a head*

*entity ($h$) and a tail entity ($t$), and a relation ($r$) connecting them. A KG $G_k$ can also be viewed as a graph $G_k = (\mathcal{V}_k, \mathcal{E}_k)$ with a set of nodes $\mathcal{V}_k$ and a set of edges $\mathcal{E}_k$.*

**Definition 3 (M-GNN)** *M-GNN is a graph encoder $f : \mathcal{M} \to \mathbb{R}^d$ that is capable of encoding a molecule graph (MG) to a vector $\mathbf{h}_{\mathrm{MG}}$.*

**Definition 4 (K-GNN)** *K-GNN is a graph encoder $g : \mathcal{K} \to \mathbb{R}^d$ that is capable of encoding the central molecule in a molecule KG sub-graph to a vector $\mathbf{h}_{\mathrm{KG}}$.*

Our GODE approach (illustrated in Figure 1) first conducts molecule-level pre-training to train an M-GNN and KG-level pre-training to train a K-GNN with a series of self-supervised tasks. Subsequently, we employ contrastive learning to enhance the alignment of molecule representations between the pre-trained M-GNN and K-GNN. Finally, we fine-tune our model for molecular property prediction tasks. We break down our approach in the subsequent sections step by step.

## 3.1 MOLECULE-LEVEL PRE-TRAINING

Given a molecular graph $G_m = (\mathcal{V}_m, \mathcal{E}_m)$, we employ the GNN encoder to derive embeddings for atoms and bonds. To pre-train M-GNN, we employ two tasks described below.

(1) Node-level Contextual Property Prediction. We randomly select a node $v \in \mathcal{V}_m$ and its corresponding embedding $\mathbf{h}_v$. This embedding is then input into an output layer for predicting the contextual property. Contextual property prediction operates as a multi-class classification task. Here, the GNN's output layer computes the probability distribution for potential contextual property labels linked to node $v$. These labels originate from the statistical attributes of the sub-graph centered on $v$ (Rong et al., 2020).

(2) Graph-level Motif Prediction. The molecule graph embedding, represented as $\mathbf{h}_{\mathrm{MG}}$, is also input into an output layer. This layer predicts the presence or absence of functional group motifs, which is detected by RDKit (Landrum et al., 2013). The embedding $\mathbf{h}_{\mathrm{MG}}$ is derived by applying mean pooling to all nodes: $\mathbf{h}_{\mathrm{MG}} = \mathrm{MEAN}(\mathbf{h}_{v_1}, \mathbf{h}_{v_2}, ..., \mathbf{h}_{v_k}|v_1, v_2, ..., v_k \in \mathcal{V}_m)$, where $\mathbf{h}_{v_1}$, $\mathbf{h}_{v_2}$, ..., $\mathbf{h}_{v_k}$ are the learned node embeddings from the M-GNN's final convolutional layer. This prediction task is a multi-label classification problem, where the GNN output layer predicts a binary label vector, indicating the presence or absence of each functional group motif in $G_m$.

During training, we employ a joint loss function, as shown in Eq. 1, to optimize both the node-level contextual property prediction and the graph-level motif prediction. This loss function encourages the M-GNN to accurately predict both the contextual properties of nodes and the functional group motifs' presence or absence in the molecule graph.

$$\mathcal{L}_{\mathrm{M}} = \sum_{v}^{\mathcal{V}'_m} \log P(p_v|\mathbf{h}_v) + \sum_{j=1}^{n} y_j \log P(M_j|\mathbf{h}_{\mathrm{MG}}) + (1 - y_j) \log(1 - P(M_j|\mathbf{h}_{\mathrm{MG}})), \quad (1)$$

where $\mathcal{V}'_m$ is a set of randomly selected nodes; $p_v$ is the contextual property label for the node $v$; $n$ is the number of all possible motifs; $M_j$ is the presence of $j$-th motif.

After the molecule-level pre-training, M-GNN is able to encode a molecule to a vector $\mathbf{h}_{\mathrm{MG}}$ through mean pooling given its molecule graph.

## 3.2 KG-LEVEL PRE-TRAINING

Embedding Initialization. Prior to the K-GNN pre-training, we use knowledge graph embedding (KGE) methods (Bordes et al., 2013; Yang et al., 2014; Sun et al., 2019; Balažević et al., 2019) to initialize the node and edge embeddings with entity and relation embeddings. KGE methods capture relational knowledge behind the structure and semantics of entities and relationships in the KG. The KGE model is trained on the entire KG ($\mathcal{T}$) and learns to represent each entity and relation as continuous vectors in a low-dimensional space. The resulting embedding vectors capture the semantic meanings and relationships between entities and relations. The loss functions of KGE methods depend on the scoring functions they use. For example, TransE (Bordes et al., 2013) learns embeddings for entities and relations in a KG by minimizing the difference between the sum of the

head entity embedding ($\mathbf{e}_h$) and the relation embedding ($\mathbf{r}_r$), and the tail entity embedding ($\mathbf{e}_t$): $s(h, r, t) = -\|\mathbf{e}_h + \mathbf{r}_r - \mathbf{e}_t\|_p$, where $\|\cdot\|_p$ is the Lp norm. After training the KGE model, we obtain the entity embeddings $\mathbf{e}_v$ and relation embeddings $\mathbf{r}_e$ for each node $v$ and edge $e$ in the KG, providing a strong starting point.

Sub-graph Extraction. for the central molecule is a crucial step in KG-level pre-training. Inspired by the work of G-Meta (Huang & Zitnik, 2020), we extract the sub-graph of each molecule to learn transferable knowledge from its surrounding nodes/edges in the biochemical KG. Specifically, for each central molecule, we extract a $\kappa$-hop sub-graph from the entire KG to capture its local neighborhood information. Given a molecule $m_i$, we first find its corresponding node $v_i$ in the KG, $G_k = (\mathcal{V}_k, \mathcal{E}_k)$. We then iteratively extract a neighborhood sub-graph $\mathcal{N}_k(v_i, h)$ of depth $h$ ($1 \leq h \leq \kappa$), centered at node $v_i$. The depth parameter $h$ determines the number of edge traversals to include in the sub-graph. To avoid over-smoothing, we stop the expansion of a graph branch when reaching a non-molecule node. Formally, the sub-graph extraction process is defined as follows. Let $\mathcal{N}_k(v, 0)$ be a single node $v$. For $h > 0$, $\mathcal{N}_k(v, h)$ is defined recursively as:

$$\mathcal{N}_k(v, h) = \{v\} \cup \bigcup_{u \in \mathcal{N}_k(v, h-1)} \{u\} \cup \bigcup_{u \in \mathcal{M}} \{w : (u, w) \in \mathcal{E}_k\}, \tag{2}$$

where $u$ denotes the set of neighboring nodes of $v$ in the sub-graph $\mathcal{N}_k(v, h-1)$, and $w : (u, w) \in \mathcal{E}_k$ represents the set of nodes that share an edge with $u \in \mathcal{M}$ in the original KG $G_k$ where $\mathcal{M}$ is the set of molecule nodes. We define The $\kappa$-hop sub-graph for molecule $m$ is given by $G_{\text{sub}(m,\kappa)} = (\mathcal{V}_{\text{sub}(m,\kappa)}, \mathcal{E}_{\text{sub}(m,\kappa)}) = \mathcal{N}_k(c, \kappa)$ where $c$ is the corresponding node of $m$ in $G_{\text{sub}(m,\kappa)}$.

We set three tasks for the KG-level pre-training as shown in module ii of Figure 1:

(1) Edge Prediction, a multi-class classification task aiming at correctly predicting the edge type between two nodes:

(2) Node Prediction, a multi-class classification task predicting the category of a node in $G_{\text{sub}(m,\kappa)}$;

(3) Node-level Motif Prediction, a multi-label classification task predicting the motif of the central molecule node $c$ in $G_{\text{sub}(m,\kappa)}$. The motif labels are created by RDKit.

The following loss function is used to pre-train K-GNN:

$$\mathcal{L}_{\text{K}} = -\left[\lambda_{\text{edge}} \underbrace{\sum_{(u,v)}^{\mathcal{E}_{\text{sub}(m,\kappa)}} \log P((u,v)'|\mathbf{h}_u \oplus \mathbf{h}_v)}_{\text{edge prediction}} + \lambda_{\text{node}} \underbrace{\sum_{v}^{\mathcal{V}_{\text{sub}(m,\kappa)}} [\log P(v'|\mathbf{h}_v)]}_{\text{node prediction}}\right.$$
$$\left. + \lambda_{\text{mot}} \underbrace{\sum_{j=1}^{n} [y_j \log P(M_j|\mathbf{h}_c) + (1 - y_j) \log(1 - P(M_j|\mathbf{h}_c))]}_{\text{motif prediction}}, \right. \tag{3}$$

where the first term $(u, v)'$ is the label of edge between the nodes $u$ and $v$. $v'$ is the label of node $v$, $\oplus$ denotes the embedding concatenation. $y_j$ is binary indicator, $\log P(M_j|\mathbf{h}_c)$ is the predicted probability of central molecule $c$ has the $j$-th functional group motif $M_j$ given its embedding $\mathbf{h}_c$. $\lambda_{\text{edge}}$, $\lambda_{\text{mot}}$, and $\lambda_{\text{mol}}$ are hyperparameters balancing the importance of different tasks.

After the KG-level pre-training, K-GNN can encode a molecule to a vector $\mathbf{h}_{\text{KG}}$ given its surrounding nodes in the KG sub-graph.

### 3.3 CONTRASTIVE LEARNING

Inspired by the success of previous works (Radford et al., 2021; Seidl et al., 2023; Sanchez-Fernandez et al., 2022) that apply contrastive learning to transfer knowledge across different modalities, we follow their steps using InfoNCE as the loss function to conduct contrastive learning between molecule graph and KG sub-graph. We construct the training set $\mathcal{D} = \mathcal{D}^+ \cup \mathcal{D}^- = \{(m_i, s_i), y_i\}_N$, where $\mathcal{D}^+ = \{(m_i, G_{\text{sub}(m_i,\kappa)}), y_i = 1\}_{N_p}$ is a set of positive samples and $\mathcal{D}^- = \{(m_i, G_{\text{sub}(m_j,\kappa)})_{j \neq i}, y_i = 0\}_{N-N_p}$ is a set of negative samples. To make the task more

Table 1: **Overview of MolKG**, a biochemical dataset we construct from PubChemRDF and PrimeKG.

| **# Triples**: 2523867 | **# Entities**: 184819 | **# Relations**: 39 | **# Entity Types**: 7 | **# Molecules**: 65454 |
|---|---|---|---|---|

**Entity Types**
*molecule,   gene/protein,   disease,   effect/phenotype,   drug,   pathway,   value*

**Relations**
*drug_protein,   contraindication,   indication,   off-label use,   drug_drug,   drug_effect,   defined_bond_stereo_count,   tpsa,   rotatable_bond_count, xlogp3-aa,   structure_complexity,   covalent_unit_count,   defined_atom_stereo_count,   molecular_weight,   hydrogen_bond_donor_count, undefined_bond_stereo_count,   isotope_atom_count,   exact_mass,   mono_isotopic_weight,   total_formal_charge,   hydrogen_bond_acceptor_count, non-hydrogen_atom_count,   tautomer_count,   undefined_atom_stereo_count,   xlogp3,   cooccurence_molecule_molecule,   cooccurence_molecule_disease, cooccurence_molecule_gene/protein,   neighbor_2d,   neighbor_3d,   has_same_connectivity,   has_component,   has_isotopologue,   has_parent, has_stereoisomer,   to_drug,   closematch,   type,   in_pathway*

challenging, we further divide $\mathcal{D}^-$ into $\mathcal{D}^-_{rand}$, and $\mathcal{D}^-_{nbr}$, which are (1) randomly sampled from all negative molecule-centric KG sub-graphs , and (2) sampled from the sub-graphs of the neighbor molecule nodes connected to the positive molecule node, respectively. The loss is defined as:

$$\mathcal{L}_{\text{InfoNCE}} = -\frac{1}{N} \sum_{i=1}^{N} \left[ y_i \log(\text{sim}(f(m_i), g(s_i))) + (1 - y_i) \log(1 - \text{sim}(f(m_i), g(s_i))) \right], \quad (4)$$

where $\text{sim}(f(m_i), g(s_i))) = \frac{\exp\left(\tau^{-1}\mathbf{h}^{\text{T}}_{\text{MG}(i)}\mathbf{h}_{\text{KG}(i)}\right)}{\exp\left(\tau^{-1}\mathbf{h}^{\text{T}}_{\text{MG}(i)}\mathbf{h}_{\text{KG}(i)}+1\right)}$, $y_i$ is the binary label, $m_i$ and $s_i$ are the paired MG and KG sub-graph in the training data, $\tau^{-1}$ is the inverse temperature.

### 3.4 FINE-TUNING FOR DOWNSTREAM TASKS

Upon completing molecule- and KG-level pre-training combined with contrastive learning, we obtain two GNN encoders, $f$ and $g$, which respectively encode molecules and KG sub-graphs into vectors. We further employ RDKit, in line with approaches from (Rong et al., 2020; Fang et al., 2023; Wu et al., 2018; Yang et al., 2019) to extract additional molecule-level features $\mathbf{h}_{\text{f}}$. A joint representation is formed by $\mathbf{h}_{\text{joint}} = \mathbf{h}_{\text{MG}} \oplus \mathbf{h}_{\text{f}} \oplus \mathbf{h}_{\text{KG}}$, with $\oplus$ representing concatenation. This representation is then utilized to predict the target property $y$ using a multi-layer perception (MLP) with an appropriate activation function. For multi-label classification, we employ binary cross-entropy loss with sigmoid activation, and for regression, we use Mean Squared Error (MSE) loss.

## 4 EXPERIMENTS

### 4.1 EXPERIMENTAL SETTING

**Data Sources.** (1) Molecule-level pre-training data: The pre-training data for our molecule-level M-GNN is derived from the same unlabelled dataset of 11 million molecules utilized by GROVER. This dataset encompasses sources such as ZINC15 (Sterling & Irwin, 2015) and ChEMBL (Gaulton et al., 2012). We randomly split this dataset into two subsets with a 9:1 ratio for training and validation. (2) Knowledge graph-level pre-training data: For the KG-level GNN (K-GNN), we select knowledge graph triples related to the molecules from PubChemRDF and PrimeKG. These include various subdomains and properties from PubChemRDF, as well as 3-hop sub-graphs for all 7957 drugs from PrimeKG. We show an overview of the dataset in Table 1. The dataset is divided into training and validation sets with a 9:1 ratio. The detailed construction of the dataset is outlined in the appendix. (3) Contrastive learning data: we set the negative/positive sample ratio as $\alpha = \frac{|\mathcal{D}^-|}{|\mathcal{D}^+|} = 32$ and retain a $1:1$ ratio for $\mathcal{D}^-_{rand} : \mathcal{D}^-_{nbr}$. Training and validation samples are in a $0.95 : 0.05$ ratio. (4) Downstream task datasets: The effectiveness of our model is tested utilizing the comprehensive MoleculeNet dataset (Wu et al., 2018; Huang et al., 2021)[1], which contains 6 classification and 5 regression datasets for molecular property prediction. We place detailed descriptions of these datasets in the Appendix. To fine-tune the model, we calculate the mean and standard deviation of the ROC-AUC for classification tasks and RMSE/MAE for regression tasks. Scaffold[2] splitting with three random seeds was employed with a training/validation/testing ratio of 8:1:1 across all datasets, aligning with methodologies employed in previous studies (Rong et al., 2020; Fang et al., 2023).

**Implementation.** For molecule-level pre-training, we employ GROVER (Rong et al., 2020), and for KG-level pre-training, we utilize GINE (Hu et al., 2019). TransE initializes the KG embeddings over

---

[1]`https://moleculenet.org/datasets-1`

[2]Scaffolds are molecular substructures; different scaffolds typically confer different chemical properties.

Table 2: **ROC-AUC performance on six *classification* benchmarks (*higher is better*).** We report the mean and standard deviation. Top-3 and top-1 results are highlighted in **bold** and **bold red**, respectively. We highlight the backbone model, and the models that apply the backbone. Table split: Non-KG and KG-based methods.

| Dataset | BBBP | SIDER | ClinTox | BACE | Tox21 | ToxCast |
|---|---|---|---|---|---|---|
| # Molecules | 2039 | 1427 | 1478 | 1513 | 7831 | 8575 |
| # Tasks | 1 | 27 | 2 | 1 | 12 | 617 |
| GCN (Kipf & Welling, 2016) | $71.8 \pm 0.9$ | $53.6 \pm 0.3$ | $62.5 \pm 2.8$ | $71.6 \pm 2.0$ | $70.9 \pm 0.3$ | $65.0 \pm 6.1$ |
| GIN (Xu et al., 2018) | $65.8 \pm 4.5$ | $57.3 \pm 1.6$ | $58.0 \pm 4.4$ | $70.1 \pm 5.4$ | $74.0 \pm 0.8$ | $66.7 \pm 1.5$ |
| SchNet (Schütt et al., 2017) | $84.8 \pm 2.2$ | $54.5 \pm 3.8$ | $71.7 \pm 4.2$ | $76.6 \pm 1.1$ | $76.6 \pm 2.5$ | $67.9 \pm 2.1$ |
| MPNN (Gilmer et al., 2017) | $91.3 \pm 4.1$ | $59.5 \pm 3.0$ | $87.9 \pm 5.4$ | $81.5 \pm 4.4$ | $80.8 \pm 2.4$ | $69.1 \pm 1.3$ |
| DMPNN (Yang et al., 2019) | $91.9 \pm 3.0$ | $63.2 \pm 2.3$ | $89.7 \pm 4.0$ | $85.2 \pm 5.3$ | $\mathbf{82.6 \pm 2.3}$ | $71.8 \pm 1.1$ |
| MGCN (Lu et al., 2019) | $85.0 \pm 6.4$ | $55.2 \pm 1.8$ | $63.4 \pm 4.2$ | $73.4 \pm 3.0$ | $70.7 \pm 1.6$ | $66.3 \pm 0.9$ |
| N-GRAM (Liu et al., 2019) | $91.2 \pm 1.3$ | $63.2 \pm 0.5$ | $85.5 \pm 3.7$ | $87.6 \pm 3.5$ | $76.9 \pm 2.7$ | - |
| HU. et.al (Hu et al., 2019) | $70.8 \pm 1.5$ | $62.7 \pm 0.8$ | $72.6 \pm 1.5$ | $84.5 \pm 0.7$ | $78.7 \pm 0.4$ | $65.7 \pm 0.6$ |
| GROVER$_{\text{Large, GTrans}}$ (Rong et al., 2020) | $86.2 \pm 3.9$ | $57.6 \pm 1.6$ | $74.7 \pm 4.4$ | $82.5 \pm 4.4$ | $76.9 \pm 2.3$ | $66.7 \pm 2.6$ |
| MGSSL (Zhang et al., 2021) | $70.5 \pm 1.1$ | $64.1 \pm 0.7$ | $80.7 \pm 2.1$ | $79.7 \pm 0.8$ | $76.4 \pm 0.4$ | $64.1 \pm 0.7$ |
| MolCLR (Wang et al., 2021b) | $73.3 \pm 1.0$ | $61.2 \pm 3.6$ | $89.8 \pm 2.7$ | $82.8 \pm 0.7$ | $74.1 \pm 5.3$ | $65.9 \pm 2.1$ |
| MolCLR$_{\text{GTrans}}$ (Wang et al., 2021b) | $76.7 \pm 2.2$ | $63.3 \pm 2.5$ | $89.3 \pm 3.1$ | $87.7 \pm 1.8$ | $80.2 \pm 3.2$ | $70.4 \pm 2.1$ |
| KGE_NFM (Ye et al., 2021) w/ our MolKG | $92.4 \pm 2.4$ | $\mathbf{65.3 \pm 1.4}$ | $87.3 \pm 2.0$ | $78.1 \pm 2.1$ | $79.8 \pm 3.3$ | $\mathbf{72.6 \pm 1.8}$ |
| KANO$_{\text{CMPNN}}$ (Fang et al., 2022a) | $\mathbf{92.6 \pm 1.8}$ | $\mathbf{65.5 \pm 1.6}$ | $\mathbf{92.9 \pm 1.1}$ | $\mathbf{90.7 \pm 3.1}$ | $81.8 \pm 1.1$ | $\mathbf{72.5 \pm 1.9}$ |
| KANO$_{\text{GTrans}}$ (Fang et al., 2023) | $\mathbf{93.7 \pm 2.3}$ | $63.8 \pm 1.2$ | $\mathbf{93.6 \pm 0.7}$ | $\mathbf{90.4 \pm 1.5}$ | $\mathbf{81.2 \pm 1.8}$ | $\mathbf{72.5 \pm 1.5}$ |
| GODE (ours) | $\mathbf{\color{red}94.5 \pm 1.9}$ | $\mathbf{\color{red}67.2 \pm 1.4}$ | $\mathbf{\color{red}94.1 \pm 2.9}$ | $\mathbf{\color{red}91.8 \pm 2.2}$ | $\mathbf{\color{red}84.3 \pm 1.2}$ | $\mathbf{\color{red}73.0 \pm 0.9}$ |

Table 3: **RMSE (for FreeSolv, ESOL, Lipophilicity) and MAE (for QM7/8) performance on five *regression* benchmarks (*lower is better*).** Top-3 and top-1 results are highlighted in **bold** and **bold red**, respectively. We highlight the backbone model, and the models that apply the backbone.

| Datasets | FreeSolv | ESOL | Lipophilicity | QM7 | QM8 |
|---|---|---|---|---|---|
| # Molecules | 642 | 1128 | 4200 | 6830 | 21786 |
| # Tasks | 1 | 1 | 1 | 1 | 12 |
| GCN (Kipf & Welling, 2016) | $2.870 \pm 0.140$ | $1.430 \pm 0.050$ | $0.712 \pm 0.049$ | $122.9 \pm 2.2$ | $0.037 \pm 0.001$ |
| GIN (Xu et al., 2018) | $2.765 \pm 0.180$ | $1.452 \pm 0.020$ | $0.850 \pm 0.071$ | $124.8 \pm 0.7$ | $0.037 \pm 0.001$ |
| SchNet (Schütt et al., 2017) | $3.215 \pm 0.755$ | $1.045 \pm 0.064$ | $0.909 \pm 0.098$ | $74.2 \pm 6.0$ | $0.020 \pm 0.002$ |
| MPNN (Gilmer et al., 2017) | $1.621 \pm 0.952$ | $1.167 \pm 0.430$ | $\mathbf{0.672 \pm 0.051}$ | $111.4 \pm 0.9$ | $\mathbf{0.015 \pm 0.001}$ |
| DMPNN (Yang et al., 2019) | $1.673 \pm 0.082$ | $1.050 \pm 0.008$ | $0.683 \pm 0.016$ | $103.5 \pm 8.6$ | $\mathbf{0.016 \pm 0.001}$ |
| MGCN (Lu et al., 2019) | $3.349 \pm 0.097$ | $1.266 \pm 0.147$ | $1.113 \pm 0.041$ | $77.6 \pm 4.7$ | $0.022 \pm 0.002$ |
| N-GRAM (Liu et al., 2019) | $2.512 \pm 0.190$ | $1.100 \pm 0.160$ | $0.876 \pm 0.033$ | $125.6 \pm 1.5$ | $0.032 \pm 0.003$ |
| HU. et.al (Hu et al., 2019) | $2.764 \pm 0.002$ | $1.100 \pm 0.006$ | $0.739 \pm 0.003$ | $113.2 \pm 0.6$ | $0.022 \pm 0.001$ |
| GROVER$_{\text{Large, GTrans}}$ (Rong et al., 2020) | $2.445 \pm 0.761$ | $1.028 \pm 0.145$ | $0.890 \pm 0.050$ | $95.3 \pm 5.6$ | $0.020 \pm 0.003$ |
| MolCLR (Wang et al., 2021b) | $2.301 \pm 0.247$ | $1.113 \pm 0.023$ | $0.789 \pm 0.009$ | $90.0 \pm 1.7$ | $0.019 \pm 0.013$ |
| MolCLR$_{\text{GTrans}}$ (Wang et al., 2021b) | $2.124 \pm 0.223$ | $0.982 \pm 0.109$ | $0.767 \pm 0.064$ | $88.9 \pm 4.8$ | $0.018 \pm 0.002$ |
| KGE_NFM (Ye et al., 2021) w/ our MolKG | $1.942 \pm 0.441$ | $1.027 \pm 0.201$ | $0.877 \pm 0.071$ | $87.6 \pm 3.2$ | $\mathbf{0.016 \pm 0.001}$ |
| KANO$_{\text{CMPNN}}$ (Fang et al., 2022a) | $\mathbf{1.320 \pm 0.244}$ | $\mathbf{0.902 \pm 0.104}$ | $\mathbf{\color{red}0.641 \pm 0.012}$ | $\mathbf{66.5 \pm 3.7}$ | $\mathbf{\color{red}0.014 \pm 0.001}$ |
| KANO$_{\text{GTrans}}$ (Fang et al., 2023) | $\mathbf{1.443 \pm 0.315}$ | $\mathbf{0.914 \pm 0.092}$ | $\mathbf{0.651 \pm 0.018}$ | $63.6 \pm 4.1$ | $\mathbf{\color{red}0.014 \pm 0.002}$ |
| GODE (ours) | $\mathbf{\color{red}1.129 \pm 0.314}$ | $\mathbf{\color{red}0.785 \pm 0.128}$ | $0.743 \pm 0.043$ | $\mathbf{\color{red}57.2 \pm 3.0}$ | $\mathbf{\color{red}0.014 \pm 0.001}$ |

a span of 10 epochs. Our settings include $\lambda_{\text{edge}} = 1.5$, $\lambda_{\text{mot}} = 1.8$, and $\lambda_{\text{node}} = 1.5$. Both M-GNN and K-GNN have a hidden size of 1,200. We adopt a temperature $\tau = 1.0$ for contrastive learning. Early stopping is anchored to validation loss. During fine-tuning, embeddings from K-GNN remain fixed, updating only the parameters of M-GNN. We use Adam optimizer with the Noam learning rate scheduler (Vaswani et al., 2017). All tests are performed on a setup featuring two AMD EPYC 7513 32-Core Processors, 528GB RAM, 8 NVIDIA RTX A6000 GPUs, and CUDA 11.7.

**Baselines.** We compare our proposed model, GODE, with several popular baselines in molecular property prediction tasks. These baselines include GCN (Kipf & Welling, 2016), GIN (Xu et al., 2018), SchNet (Schütt et al., 2017), MPNN (Gilmer et al., 2017), DMPNN (Yang et al., 2019), MGCN (Lu et al., 2019), N-GRAM (Liu et al., 2019), Hu et al (Hu et al., 2019), GROVER (Rong et al., 2020), MGSSL (Zhang et al., 2021), KGE_NFM (Ye et al., 2021), MolCLR (Wang et al., 2021b), and KANO (Fang et al., 2023).

## 4.2 PERFORMANCE IN MOLECULE PROPERTY PREDICTION

Tables 2 and 3 present comparative performance metrics for classification and regression tasks, respectively. It is clear from the data that our proposed method, GODE, consistently outperforms the baseline models in most tasks. Specifically, in classification tasks, GODE achieves SOTA results across all tasks. Amongst the competitors, KANO stands out, consistently showcasing performance close to our method. Intriguingly, KANO, as a knowledge-driven model, augments molecular struc-

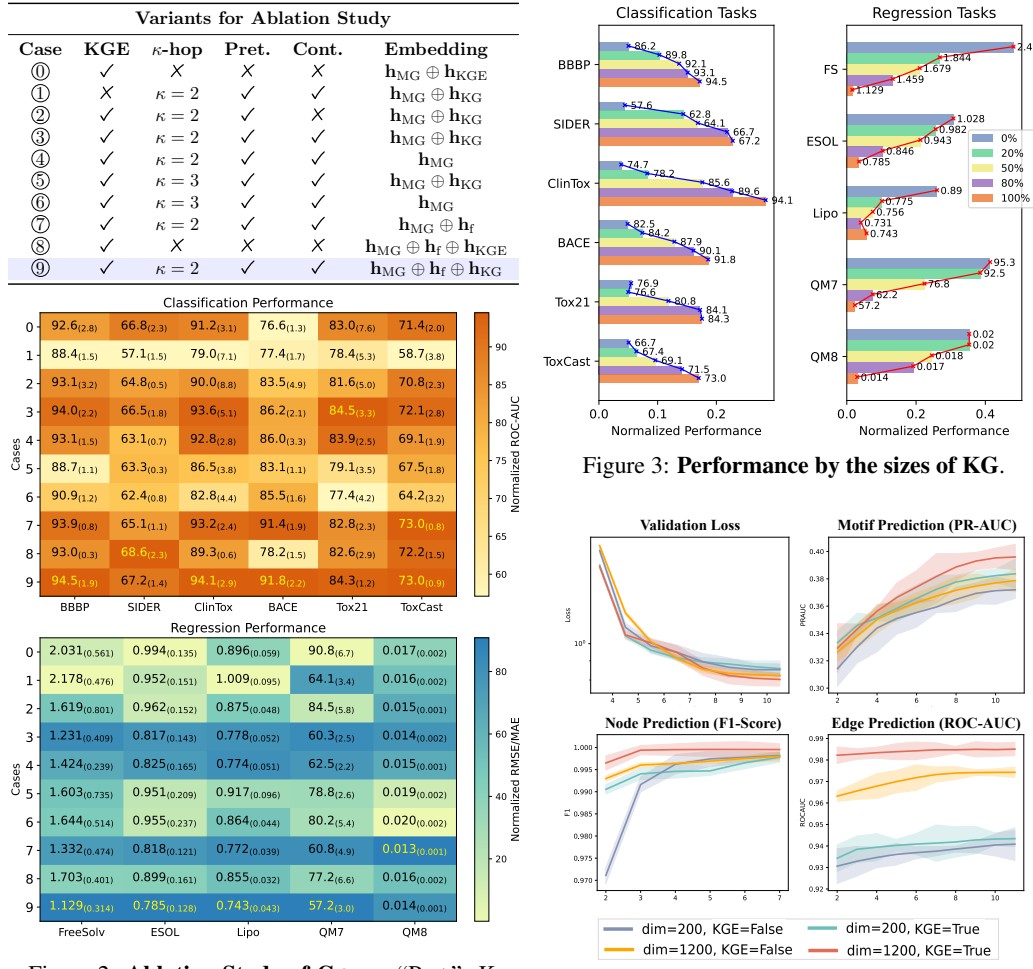

**Variants for Ablation Study**

| Case | KGE | $\kappa$-hop | Pret. | Cont. | Embedding |
|---|---|---|---|---|---|
| ⓪ | ✓ | ✗ | ✗ | ✗ | $\mathbf{h}_{MG} \oplus \mathbf{h}_{KGE}$ |
| ① | ✗ | $\kappa = 2$ | ✓ | ✓ | $\mathbf{h}_{MG} \oplus \mathbf{h}_{KG}$ |
| ② | ✓ | $\kappa = 2$ | ✓ | ✗ | $\mathbf{h}_{MG} \oplus \mathbf{h}_{KG}$ |
| ③ | ✓ | $\kappa = 2$ | ✓ | ✓ | $\mathbf{h}_{MG} \oplus \mathbf{h}_{KG}$ |
| ④ | ✓ | $\kappa = 2$ | ✓ | ✓ | $\mathbf{h}_{MG}$ |
| ⑤ | ✓ | $\kappa = 3$ | ✓ | ✓ | $\mathbf{h}_{MG} \oplus \mathbf{h}_{KG}$ |
| ⑥ | ✓ | $\kappa = 3$ | ✓ | ✓ | $\mathbf{h}_{MG}$ |
| ⑦ | ✓ | $\kappa = 2$ | ✓ | ✓ | $\mathbf{h}_{MG} \oplus \mathbf{h}_{f}$ |
| ⑧ | ✓ | ✗ | ✗ | ✗ | $\mathbf{h}_{MG} \oplus \mathbf{h}_{f} \oplus \mathbf{h}_{KGE}$ |
| ⑨ | ✓ | $\kappa = 2$ | ✓ | ✓ | $\mathbf{h}_{MG} \oplus \mathbf{h}_{f} \oplus \mathbf{h}_{KG}$ |

Figure 2: **Ablation Study of GODE.** "Pret.": K-GNN pre-training. "Cont": contrastive learning. "Embedding": input to MLP for fine-tuning. The best setting is shaded. The best result for each task is highlighted.

Figure 3: **Performance by the sizes of KG.**

Figure 4: **Performance of KG-level (K-GNN) pre-training tasks by time.** We report the means and standard deviation based on five runs with different random seeds.

tures by integrating information about chemical elements from its ElementKG. This underlines the substantial advantage of leveraging external knowledge in predicting molecular properties. On the regression front, GODE attains best results in 4 out of 5 tasks. This consistent high performance, irrespective of the nature of the task, underscores our model's adaptability and reliability. Cumulatively, there is a relative improvement of 23.6% across all tasks with our approach (12.7% for classification and 34.4% for regression tasks). When compared with the SOTA model, KANO, GODE records improvements of 2.1% and 4.2% for classification and regression tasks, respectively. To analyze the effects of GODE's variants, we conduct ablation studies in Figure 2, which are discussed as follows.

**Effect of the Integration of MolKG.** To assess the impact of integrating our molecule-centric KG - MolKG, into molecule property prediction, we juxtapose Case ⑧ with our backbone M-GNN model, GROVER. Specifically, Case ⑧ melds GROVER ($\mathbf{h}_{MG} \oplus \mathbf{h}_{f}$) with the static KG embedding ($\mathbf{h}_{KGE}$), which is trained using the KGE method. Our observations indicate that infusing the KG boosts performance across all tasks, resulting in a noteworthy 14.3% overall enhancement. Moreover, when all variants of GODE are deployed (as in Case ⑨), a significant uplift of 23.2% in performance over GROVER is realized. We further study the relationship between KG size and task performance (as per Case ⑨) by randomly sampling triples (20%, 50%, and 80%) for each relational type. Figure 3 discerns a consistent upward trajectory in performance commensurate with the growth of the KG size. This trend underscores the merit of a more comprehensive knowledge repository. One of our future works is to augment MolKG with additional molecules to further refine K-GNN pre-training.

**Effect of KG-level Pre-training and Contrastive Learning.** Through a side-by-side comparison of Cases ⓪, ②, and ③, we discern the value of K-GNN pre-training and contrastive learning.

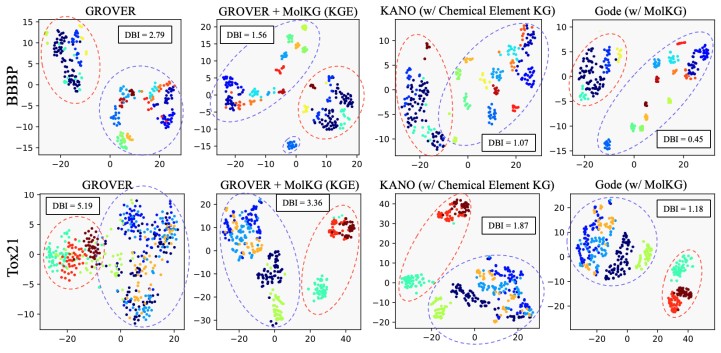

Figure 5: **t-SNE visualization of molecule embeddings across two tasks**. Each color represents a unique scaffold (molecule substructure). We compare the embeddings from GROVER, GROVER augmented with static KG embeddings from our MolKG, KANO, and GODE. The clustering quality is assessed using the DB index.

Standalone K-GNN pre-training (Case ②) yields a modest boost of 4.5%, with a particularly slight edge in classification tasks at 0.1%. However, when paired with contrastive learning and leveraging both $\mathbf{h}_{\text{MG}}$ and $\mathbf{h}_{\text{KG}}$ for fine-tuning, as in Case ③, the surge in performance is notable, reaching an overall enhancement of 13.6% over the baseline Case ⓪. A testament to the effectiveness of this approach can be seen in the BBBP dataset. The molecule acetylsalicylate, better known as aspirin, posed a prediction challenge to both our M-GNN model and the methods in Cases ⓪ and ②. Yet, when Case ③ employed relational knowledge from its KG subgraph (e.g., [*acetylsalicylate, indication, neurological conditions*]) alongside contrastive learning, it managed to make accurate predictions. This example underscores the pivotal role of contrastive learning in refining molecular property predictions.

**Effect of Embedding Initialization.** Figure 4 illustrates the pivotal role of KGE embedding initialization in augmenting the efficacy of K-GNN pre-training tasks (see Eq. 3). This advantage manifests as enhanced task performance and consistently diminished validation loss, signifying sharper predictions. The data also indicates a direct relationship between embedding dimensionality and pre-training quality: larger dimensions consistently yield superior results. Further, Figure 2 emphasizes that the inclusion of KGE embeddings, as in Case ③, consistently surpasses scenarios excluding them, such as Case ①. This accentuates the significance of KGE embedding initialization in GODE.

**Efficacy of Knowledge Transfer.** The influence of contrastive learning in transferring domain knowledge from the biochemical KG to the molecular representation $\mathbf{h}_{\text{MG}}$ is discerned by examining Cases ③ to ⑥ and contrasting GROVER with Cases ⑦ and ⑨. Notably, while the M-GNN embeddings of GODE (represented by Cases ④ and ⑥) do not quite surpass the bi-level concatenated embeddings (Cases ③ and ⑤), they come notably close. More compelling is Case ⑦, which parallels Case ⑨ and outperforms GROVER by a striking 21.0% (with 12.0% in classification and 30.1% in regression). For particular tasks, such as ToxCase and QM8, it even achieves slightly better results than Case ⑨. The distinguishing feature of Case ⑦ that provides an edge over GROVER is its enriched $\mathbf{h}_{\text{MG}}$, an enhancement absent in GROVER. This underscores GODE's prowess in knowledge transfer to molecular representations and its resilience in the absence of a molecule-related KG.

**Insights from Embedding Visualization.** In the t-SNE visualization presented in Figure 5, the GROVER embeddings highlight molecules from varying scaffolds intermingling, signaling a significant avenue for refinement. Particularly in the Tox21 task, these embeddings appear sparse. When enhanced with MolKG's static KGE, there is a noticeable delineation of clusters, reflecting the constructive influence of our MolKG in integrating biochemical nuances into molecular representations. Nonetheless, a residual overlap of molecules from different scaffolds still persists. Progressing to the GODE visualization, the clusters exhibit further refinement, achieving pronounced distinctiveness with minimal scaffold overlap, outperforming KANO (with Chemical Element KG), and securing the lowest Davies–Bouldin (DB) index, which underscores the effectiveness of GODE with MolKG.

## 5 CONCLUSION

We presented GODE, a framework employing bi-level self-supervised pre-training and contrastive learning to refine molecule representations using biochemical domain knowledge. Empirical evaluations confirmed its efficacy in molecular property predictions. Moving forward, we plan to broaden the scope of MolKG to cover diverse molecules and their multi-domain knowledge. Our focus will also be on identifying key knowledge aspects that optimize molecule representation. Ultimately, our work sets a foundation for progress in drug discovery.

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

**Contents of Appendix**

## A  BROADER IMPACT

The development of GODE offers a significant advance in the realm of molecular representation learning. Its broader impacts can be summarized as follows:

**Enhanced Drug Discovery** By providing a robust representation of molecules enhanced by knowledge, GODE can potentially accelerate drug discovery processes. This could lead to faster identification of potential drug candidates and reduce the time and cost associated with the introduction of new drugs into the market.

**Interdisciplinary Applications** The fusion of molecular structures with knowledge graphs can be applied beyond the realm of molecular biology. This approach can be extended to other scientific domains where entities have both intrinsic structures and are part of larger networks.

**Potential Ethical Considerations** As with any predictive model, there is a need to ensure that the data used is unbiased and representative. Misrepresentations or biases in the knowledge graph or molecular data can lead to skewed predictions, which could have implications in real-world applications, especially in drug development.

## B  KNOWLEDGE GRAPH (MOLKG) CONSTRUCTION AND PROCESSING

The construction of our molecule-centric knowledge graph - MolKG, involved a comprehensive data retrieval process of knowledge graph triples relevant to molecules. We retrieve the data from two distinguished sources: PubChemRDF[3] (Fu et al., 2015) and PrimeKG (Chandak et al., 2023a). From PubChemRDF, we concentrated on triples from six specific subdomains:

- *Compound*: This encompasses compound-specific relation types such as *parent compound*, *component compound*, and *compound identity group*.

- *Cooccurrence*: This domain captures triples like *compound-compound*, *compound-disease*, and *compound -gene* co-occurrences. By ranking co-occurrences based on their scores, we selected the top 5 compounds, diseases, and genes for each molecule, resulting in at most 15 co-occurred entities per molecule.

- *Descriptor*: This domain details explicit molecular properties including *structure complexity*, *rotatable bond*, and *covalent unit count*.

- *Neighbors*: Represents the top $N$ molecules similar in 2D and 3D structures. For our dataset, we integrated the top 3 similar molecules from both 2D and 3D structures for each molecule.

- *Component*: Associates molecules with their constituent components.

- *Same Connectivity*: Showcases molecules with identical connectivity to source molecules.

From PrimeKG, we pursued a rigorous extraction technique, deriving 3-hop sub-graphs for all 7,957 drugs, regarded as molecules, from the entirety of the knowledge graph. Consistency and accuracy in data handling were paramount. We utilized recognized information retrieval tools[45] to bridge various representations and coding paradigms for identical molecular entities. Compound ID (CID) served as our go-to medium for molecular conversions across the two knowledge graphs.

Lastly, within our assembled knowledge graph, entities identified as "value" are normalized to (1, 10). Subsequently, we classified these entities, ensuring a maximum class count of 10.

We attached the entire MolKG dataset (as "*gode_data/data_process/KG_processed.csv*") and the detailed processing scripts for its construction (in "*gode_data/dataset_construction/*") as supplemental material.

---

[3]https://pubchem.ncbi.nlm.nih.gov/docs/rdf-intro
[4]https://pubchem.ncbi.nlm.nih.gov/docs/pug-rest
[5]https://www.ncbi.nlm.nih.gov/home/develop/api/

# C  DATASETS OF DOWNSTREAM TASKS

In this section, we introduce the datasets/tasks we used for evaluation.

## C.1  CLASSIFICATION DATASETS

Table 4: Description of Classification Datasets

| Dataset | # Molecules | # Tasks | Description |
| --- | --- | --- | --- |
| **BBBP** (Martins et al., 2012) | 2039 | 1 | The Blood-Brain Barrier Penetration (BBBP) dataset aids drug discovery, especially for neurological disorders. It characterizes a compound's ability to cross the blood-brain barrier, influencing treatment efficacy for brain disorders. |
| **SIDER** (Kuhn et al., 2016) | 1427 | 27 | The Side Effect Resource (SIDER) provides adverse effects data of marketed medications. This is crucial for pharmacovigilance, enabling potential side effects predictions of new compounds based on molecular properties. |
| **ClinTox** (Gayvert et al., 2016) | 1478 | 2 | ClinTox compares drugs that gained FDA approval versus those rejected due to toxic concerns. This assists researchers in anticipating toxicological profiles of new compounds. |
| **BACE** (Subramanian et al., 2016) | 1513 | 1 | The BACE dataset offers insights into potential inhibitors for human $\beta$-secretase 1 (BACE-1), an enzyme linked to Alzheimer's. It's vital for neurological drug discovery targeting Alzheimer's treatments. |
| **Tox21** (Huang & Xia, 2017) | 7831 | 12 | Tox21 offers a comprehensive toxicity profile of compounds. Central to the 2014 Tox21 Data Challenge, it aims at enhancing predictions for toxic responses to ensure safer drug design. |
| **ToxCast** (Richard et al., 2016) | 8575 | 617 | ToxCast provides toxicity labels from high-throughput screenings, enabling swift evaluations and guiding early drug development stages. |

## C.2  REGRESSION DATASETS

Table 5: Description of Regression Datasets

| Dataset | # Molecules | # Tasks | Description |
| --- | --- | --- | --- |
| **FreeSolv** (Mobley & Guthrie, 2014) | 642 | 1 | A dataset that brings together information on the hydration free energy of molecules in water. The dual presence of experimental data and alchemical free energy calculations offers researchers a robust platform to understand solvation processes and predict such properties for novel molecules. |
| **ESOL** (Delaney, 2004) | 1128 | 1 | Understanding the solubility of compounds is fundamental in drug formulation and delivery. The ESOL dataset chronicles solubility attributes, providing a structured framework to predict and modify solubility properties in drug design. |
| **Lipophilicity** (Gaulton et al., 2012) | 4200 | 1 | Extracted from the ChEMBL database, this dataset focuses on a compound's affinity for lipid bilayers—a key factor in drug absorption and permeability. It provides valuable insights derived from octanol/water distribution coefficient experiments. |
| **QM7** (Blum & Reymond, 2009) | 6830 | 1 | A curated subset of GDB-13, the QM7 dataset houses details on computed atomization energies of stable, potentially synthesizable organic molecules. It provides an arena for validating quantum mechanical methods against empirical data, bridging computational studies with experimental chemistry. |
| **QM8** (Ramakrishnan et al., 2015) | 21786 | 12 | A more extensive dataset, QM8 encompasses computer-generated quantum mechanical properties. It details aspects like electronic spectra and the excited state energy of molecules, offering a robust resource for computational chemists aiming to predict or understand such attributes. |

# D  IMPLEMENTATION DETAILS

## D.1  HYPER-PARAMETERS OF GODE

We summarize our hyper-parameter study in Table 6. Following previous works (Rong et al., 2020; Fang et al., 2023), we use RDKit to extract additional features (dimension 200) of M-GNN.

Table 6: Summary of hyper-parameter study for the experimental setup. We **highlight** the best setting we used in our experiments.

| Hyper-parameter | Studied Values |
| --- | --- |
| **M-GNN** | |
| GNN model | $\text{GROVER}_{\text{w}/\{\textbf{GTransformer},\text{MPNN},\text{GIN}\}}$ |
| learning rate | 1.5e-4 |
| weight decay | 1e-7 |
| hidden dimension | {400, 800, **1200**} |
| pre-training epochs | 500 |
| dropout | {**0.1**, 0.2, 0.3} |
| attention head | 4 |
| molecule embedding (GROVER) | {atom, bond, **both**} |
| activation function | {**PReLU**, ReLU, LeakyReLU, Sigmoid} |
| **KGE** | |
| model | {**TransE**, RotatE, DistMult, TuckER} |
| learning rate | {1e-3, **1e-4**, 1e-5, 1e-6} |
| training epochs | {5, 10} |
| hidden dimension | {200, 512, **1200**} |
| **K-GNN** | |
| GNN model | {**GINE**, GAT, GCN} |
| $\kappa$-hop | {**2**, 3} |
| learning rate | {1e-3, **1e-4**, 1e-5, 1e-6} |
| weight decay | {1e-3, 1e-4, **1e-5**, 1e-6, 1e-7} |
| hidden dimension | {200, 400, 800, **1200**} |
| pre-training epochs | 100 |
| edge prediction weight $\lambda_{\text{edge}}$ | {1.0, 1.1, 1.3, **1.5**, 1.8, 2.0} |
| node prediction weight $\lambda_{\text{node}}$ | {1.0, 1.1, 1.3, **1.5**, 1.8, 2.0} |
| motif prediction weight $\lambda_{\text{mot}}$ | {1.0, 1.1, 1.3, 1.5, **1.8**, 2.0} |
| activation function | {PReLU, ReLU, **Sigmoid**, **Softmax**} |
| **Contrastive Learning** | |
| learning rate | {1e-4, **5e-4**, 1e-3, 5e-3} |
| weight decay | {**1e-3**, 1e-4, 1e-5} |
| negative sampling ratio ($\alpha$) | {4, 8, 16, **32**, 64} |
| temperature | {0.1, 0.3, 0.7, **1.0**} |
| **Fine-tuning** | |
| batch size | {4, 16, **32**, 64, 128} |
| inital learning rate (for Noam learning rate scheduler) | {1e-3, **1e5-3**, 1e-2, 1e-1, 1, 10} |
| maximum learning rate (for Noam learning rate scheduler) | 1e-3 |
| final learning rate (for Noam learning rate scheduler) | 1e-4 |
| warmup epochs | 2 |
| training epochs | 20 |
| fold number | {4, **5**, 6} |
| data splitting | scaffold splitting |
| MLP hidden size | {100, **200**, 500} |
| MLP layer number | {1, **2**, 3, 4} |
| activation function | {**ReLU**, LeakyReLU, PReLU, tanh, SELU} |

## D.2 BASELINE MODELS

In this work, we compare GODE to 13 baseline methods, including GCN (Kipf & Welling, 2016), GIN (Xu et al., 2018), SchNet (Schütt et al., 2017), MPNN (Gilmer et al., 2017), DMPNN (Yang et al., 2019), MGCN (Lu et al., 2019), N-GRAM (Liu et al., 2019), Hu et al (Hu et al., 2019), GROVER (Rong et al., 2020), MGSSL (Zhang et al., 2021), KGE_NFM (Ye et al., 2021), Mol-CLR (Wang et al., 2021b), and KANO (Fang et al., 2023).

Similar as KANO (Fang et al., 2023)[6], we reuse the results of GCN, GIN, SchNet, MGCN, N-GRAM, and HU et al. (2019) from the paper of MolCLR (Wang et al., 2021b), and reuse the results of MGSSL from its original paper. We reuse the results of MPNN, DMPNN, and MolCLR (default setup) from the paper of KANO. This ensures we compare our method to the state-of-the-art model KANO in the same setup.

We reproduced GROVER, MolCLR (with the GTransformer (Rong et al., 2020) backbone), KGE_NFM (with our MolKG), and KANO based on the source code they provided[7][8][9][10]. We reveal the implementation details as follows.

**GROVER** (Rong et al., 2020): We use the same implementation setup as described in the original paper. We use node embeddings from both node-view and edge-view GTransformers with self-attentive READOUT function for fine-tuning and property prediction. The mean value of the prediction scores from two GTransformers is used for prediction.

**MolCLR$_{\textbf{GTrans}}$** (Wang et al., 2021b): We change the backbone molecule encoder of MolCLR to GTransformer. For a fair comparison, we pre-train node-view and edge-view GTransformers (hidden dimension 1200) separately with MolCLR's contrastive learning framework. For fine-tuning and prediction, we take the same setting as GROVER we described above.

**KGE_NFM** (Ye et al., 2021): We treat this approach as a general framework fusing molecule graph with static KGE embedding (see Appedix F.2). we use node-view and edge-view pre-trained GTransformers (GROVER$_{\text{Large, GTrans}}$) as the molecule encoders and use DistMult as the static KGE method (hidden dimension 1200). For fine-tuning, we use original paper's NFM integration and update the node-view and edge-view GTransformers separately. We take the mean value of the scores from two models for the property prediction.

**KANO** (Fang et al., 2023): We implement KANO with two backbone models: CMPNN (Song et al., 2020) and GTransformer where the former is original paper's implementation, and the later is ours. For KANO$_{\text{CMPNN}}$, We keep the same setup described by the original paper and the provided code. For KANO$_{\text{GTrans}}$, we seperately train node-view and edge-view GTransformers with KANO's contrastive-based pre-training strategy, and fine-tune the pre-trained encoders with KANO's prompt-enhanced fine-tuning strategy. The mean value of prediction scores is taken for property prediction.

## E  JUSTIFICATIONS FOR GODE

In the proposed methodology, we aim to construct a powerful molecule representation via a bi-level self-supervised pre-training technique that leverages both molecular graphs (M-GNN) and Knowledge Graphs (K-GNN). To bridge these two representations and leverage the strengths of both, contrastive learning is used. To validate and support the proposed methodologies mathematically, the following are the detailed justifications and explanations:

---

[6]see "Baseline experimental setup" in "Supplementary information" on `https://www.nature.com/articles/s42256-023-00654-0`.

[7]GROVER: `https://github.com/tencent-ailab/grover`

[8]MolCLR: `https://github.com/yuyangw/MolCLR`

[9]KGE_NFM: `https://zenodo.org/records/5500305`

[10]KANO: `https://github.com/HICAI-ZJU/KANO`

### E.1 Justifications for Bi-level Pre-training

**Molecule-level Pre-training** The objective for molecule-level pre-training is to capture local atom properties (contextual property prediction) and global functional group motifs (graph-level motif prediction), as described by Eq. 1. The goal is to maximize the likelihood of the true contextual property and the motif labels given their embeddings.

The first term, $\log P(p|\mathbf{h}_v)$ in Eq. 1, is a direct log-likelihood of the true contextual property given the node embedding. Maximizing this term encourages the GNN to capture local structural information of atoms in the molecule graph. The second and third terms work in tandem for each possible motif $M_j$. If the motif $M_j$ is present (i.e., $y_j = 1$), we want to maximize $P(M_j|\mathbf{h}_{\mathrm{MG}})$. If the motif $M_j$ is absent (i.e., $y_j = 0$), we want to maximize $1 - P(M_j|\mathbf{h}_{\mathrm{MG}})$. This is achieved via maximizing the combined term $y_j \log P(M_j|\mathbf{h}_{\mathrm{MG}}) + (1 - y_j) \log(1 - P(M_j|\mathbf{h}_{\mathrm{MG}}))$.

In maximizing this loss, we ensure that our M-GNN captures both the local properties of atoms and the global properties (functional motifs) of the molecule.

**KG-level Pre-training**. The proposed loss function for the K-GNN (Eq. 3) encapsulates three main tasks: edge prediction, motif prediction, and node prediction.

The term $\lambda_{\mathrm{edge}} \sum_{(u,v)}^{\mathcal{E}_{\mathrm{sub}(m,\kappa)}} \log P((u,v)'|\mathbf{h}_u \oplus \mathbf{h}_v)$ ensures the GNN captures the relationship between two nodes. Maximizing this log-likelihood encourages the K-GNN to capture semantic meanings and relationships between entities in the KG. The motif prediction task encourages the K-GNN to capture the properties of the central molecule, much like the motif prediction in M-GNN but now in the context of a knowledge graph. By maximizing the likelihood of the motif labels, the GNN captures the molecular motifs in the context of surrounding information from the KG. The node prediction task helps the K-GNN understand the semantic roles of individual nodes in the sub-graph.

By maximizing this combined loss, the K-GNN captures edge semantics, node roles, and molecular motifs in the context of a KG.

### E.2 Justifications for Contrastive Learning

Contrastive learning inherently aligns representations originating from disparate sources. For our context, this involves the M-GNN and K-GNN systems. By ensuring that representations of the same molecule from both platforms are more proximate in latent space and concurrently distancing representations of distinct molecules, we establish an efficient mechanism for knowledge interchange.

Consider representations $\mathbf{h}_{\mathrm{MG}}$ derived from M-GNN and $\mathbf{h}_{\mathrm{KG}}$ from K-GNN. The similarity metric between these representations for a positively correlated pair is represented by $s(\mathbf{h}_{\mathrm{MG}}, \mathbf{h}_{\mathrm{KG}})$. The overarching objective of the contrastive loss is to optimize:

$$s(\mathbf{h}_{\mathrm{MG}}, \mathbf{h}_{\mathrm{KG}}) - \mathbb{E}_{\mathrm{neg}}[s(\mathbf{h}_{\mathrm{MG}}, \mathbf{h}_{\mathrm{neg}})], \tag{5}$$

where $\mathbb{E}_{\mathrm{neg}}[.]$ stands for the anticipated similarity over negatively correlated pairs. By maximizing this difference, the collective knowledge (e.g., shared motifs and properties) across both M-GNN and K-GNN becomes intrinsically woven into their respective representations.

By applying the InfoNCE loss as illustrated in Eq. 4, we ensure a refined alignment between the M-GNN and K-GNN representations. This alignment seeks to minimize the InfoNCE loss, guaranteeing that representations of identical molecules from the two models approach one another in latent space, thereby amplifying $s(\mathbf{h}_{\mathrm{MG}}, \mathbf{h}_{\mathrm{KG}})$, while representations of unalike molecules are distanced by reducing their similarity to $\mathbf{h}_{\mathrm{neg}}$.

### E.3 Fine-tuning Benefits with Contrastive Learning

When fine-tuning a model given the concatenated representations $\mathbf{h}_{\mathrm{MG}}$ and $\mathbf{h}_{\mathrm{KG}}$, the benefits of having undergone contrastive learning become evident:

**Gradient Alignment in Coherent Representations**. Consider the concatenated representation $\mathbf{h} = \mathbf{h}_{\mathrm{MG}} \oplus \mathbf{h}_{\mathrm{KG}}$. For our downstream task, let's denote the loss function as $\mathcal{L}$. The gradient of this loss with respect to our concatenated representation is $\nabla \mathcal{L} = \frac{\partial \mathcal{L}}{\partial \mathbf{h}}$.

This gradient can be decomposed into its components:

$$\nabla \mathcal{L} = \left( \frac{\partial \mathcal{L}}{\partial \mathbf{h}_{\text{MG}}}, \frac{\partial \mathcal{L}}{\partial \mathbf{h}_{\text{KG}}} \right)$$

Due to contrastive learning, the similarity between these representations is maximized, leading to aligned gradients. This means that during backpropagation, the updates to both representations are in a similar direction. The alignment can be represented using the cosine similarity between the two gradient components:

$$\text{sim} = \frac{\frac{\partial \mathcal{L}}{\partial \mathbf{h}_{\text{MG}}} \cdot \frac{\partial \mathcal{L}}{\partial \mathbf{h}_{\text{KG}}}}{\left\| \frac{\partial \mathcal{L}}{\partial \mathbf{h}_{\text{MG}}} \right\|_2 \left\| \frac{\partial \mathcal{L}}{\partial \mathbf{h}_{\text{KG}}} \right\|_2}$$

A higher similarity value indicates that the gradients are more aligned, ensuring that both representations are updated coherently.

**Consistent Information Flow**. The coherent updates ensure a consistent flow of information during backpropagation across both representations. This is crucial as it ensures that shared knowledge and patterns recognized in the molecule from both the molecular graph and the knowledge graph are reinforced together. The synchronized evolution of representations can be represented as:

$$\mathbf{h}_{\text{MG}}^{(t+1)} = \mathbf{h}_{\text{MG}}^{(t)} - \alpha \Delta \mathbf{h}_{\text{MG}}, \quad \mathbf{h}_{\text{KG}}^{(t+1)} = \mathbf{h}_{\text{KG}}^{(t)} - \alpha \Delta \mathbf{h}_{\text{KG}}$$

where $\alpha$ is the learning rate and $t$ denotes the iteration.

**Avoidance of Conflicting Gradients**. In the absence of coherent updates, there is a risk that the gradients for $\mathbf{h}_{\text{MG}}$ and $\mathbf{h}_{\text{KG}}$ might sometimes push the representations in opposite or divergent directions. This can lead to conflicting signals during training. Mathematically, if the gradients are not aligned, the angle between them $\theta$ (where $0 \leq \theta \leq \pi$) can sometimes approach $\pi$, indicating opposing directions. This can cause oscillations in the loss landscape and hinder smooth convergence.

**Enhanced Generalization**. Coherent updates also contribute to better generalization. When both representations are updated in a harmonized manner, the model is less likely to overfit to idiosyncrasies specific to one source. Instead, it focuses on patterns and features that are consistently emphasized across both sources. Mathematically, this can be visualized in the loss landscape as broader valleys (as opposed to sharp, narrow minima). Broader valleys in the loss landscape correspond to regions of the parameter space where small changes to the parameters result in small changes to the loss, indicating better generalization.

## F   COMPARISON WITH SIMILAR STUDIES

In this section, we compare the proposed GODE method with some similar studies integrating knowledge graph and molecule for molecular property predictions. Specifically, we compare with (Ye et al., 2021) and (Fang et al., 2023), as shown in Figure 6.

### F.1   KANO

KANO (Fang et al., 2023) presents Element KG, a knowledge graph that details the relational connections among chemical elements. In their approach to transferring knowledge from this KG to molecular representations, the process begins by extracting an element sub-graph tailored to a specific molecule. This sub-graph is subsequently integrated with the original molecule graph, effectively enriching the atomic structures within the molecule using the KG. For the encoding phase, they used a non-pretrained graph encoder to derive the embedding of the enhanced molecule structure. In parallel, they utilize a pre-trained graph encoder to capture the graph embedding of the molecule, with additional features sourced from RDKit. The culmination of this process is the application of contrastive learning, which aligns the embedding of the supplemented molecule with the embedding of the original molecule, which is then used to fine-tune downstream tasks. This meticulous procedure ensures an effective transfer of knowledge from elemental details to the overall molecular representation.

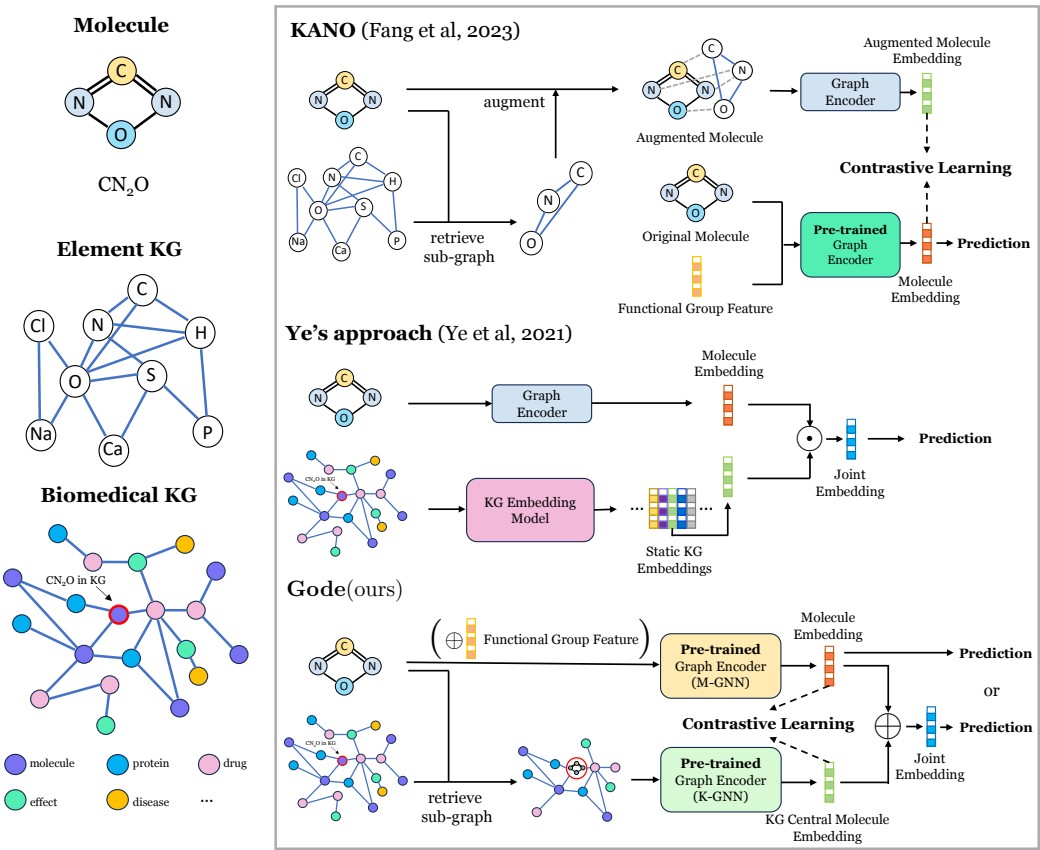

Figure 6: **An overview of the difference between GODE with similar works (Ye et al., 2021; Fang et al., 2023) leveraging both knowledge graph and molecule.** Details such as pre-training strategies or KG embedding initialization are not depicted, for clearer presentations.

### F.2 KGE_NFM

KGE_NFM (Ye et al., 2021) was initially developed for recommendation systems. However, its potential extends to predicting molecular properties, as illustrated in Figure 6. The procedure begins by obtaining embeddings for both the molecule and the biomedical knowledge graph, achieved through a molecule graph encoder and a KG embedding technique, respectively. Subsequently, element-wise multiplication is employed to combine these embeddings for predictive tasks. Notably, a primary drawback of Ye's strategy is its reliance on static, global embeddings. This can sometimes neglect the nuanced, local information pertaining to the targeted entity. Furthermore, there is a conspicuous absence of any mechanism to consolidate the same entity represented in different modalities. This omission creates a disconnect in knowledge transfer from the biomedical KG to the molecular representation.

### F.3 GODE

Gode (ours) On the other hand, our GODE methodology offers a distinct approach to integrating knowledge graphs and molecular structures for enhanced molecular property predictions. Unlike other methods, GODE directly retrieves a sub-graph tailored to the central molecule from the biochemical knowledge graph (KG). This direct retrieval ensures that the most relevant and contextual information from the KG is harnessed. GODE employs two pre-trained graph encoders, K-GNN and M-GNN, where the former is pre-trained on molecule-centric KG sub-graphs, and the latter is pre-trained on the molecule graph's structural information. An optional enhancement (Rong et al., 2020; Fang et al., 2023) to this process is the inclusion of the functional group feature, which is retrieved by RDKit. A pivotal aspect of the GODE method is the alignment process. Through the application

of contrastive learning, the representations of the same molecule, as derived from the two distinct graphs (biochemical KG and molecule graph), are meticulously aligned. This alignment ensures that the embeddings are harmonized and that there is a seamless transfer of knowledge between the two representations. In the subsequent fine-tuning stage, GODE offers flexibility. Users can either employ the concatenated embedding, which is a fusion of the outputs from M-GNN and K-GNN, or opt to use only the embedding from M-GNN. This adaptability ensures that the method can be tailored to best suit specific downstream prediction tasks, optimizing accuracy and efficiency.

# G    ADDITIONAL EXPERIMENTS

## G.1    EFFECT OF BI-LEVEL SELF-SUPERVISED PRE-TRAINING

In addition to Figure 2, we conduct more experiments to study how the pre-training of M-GNN and K-GNN affect GODE's performance.

Table 7: **Study the effect of bi-level self-supervised pre-training.** Contrastive learning is conducted between M-GNN and K-GNN no matter they are pre-trained or not. Node embedding in K-GNN are initialized by TransE. All predictions are based on the embedding $\mathbf{h}_{MG} \oplus \mathbf{h}_f \oplus \mathbf{h}_{KG}$. "M" and "K" denote M-GNN pre-training and K-GNN pre-training, respectively. $\Delta$ denotes performance gain from GODE {M:1, K:1}.

| | | **Classification Tasks** (higher is better) | | | | | | | | | | | |
|---|---|---|---|---|---|---|---|---|---|---|---|---|---|
| M | K | BBBP | $\Delta$ | SIDER | $\Delta$ | ClinTox | $\Delta$ | BACE | $\Delta$ | Tox21 | $\Delta$ | ToxCast | $\Delta$ |
| 1 | 1 | **94.5** | - | **67.2** | - | **94.1** | - | **91.8** | - | **84.3** | - | **73.0** | - |
| 0 | 1 | 92.2 | ↓2.3 | 62.6 | ↓4.6 | 89.4 | ↓4.7 | 89.8 | ↓2.0 | 80.6 | ↓3.7 | 70.8 | ↓2.2 |
| 1 | 0 | 93.2 | ↓1.3 | 66.7 | ↓0.5 | 90.7 | ↓3.4 | 81.6 | ↓10.2 | 83.1 | ↓1.2 | 71.9 | ↓1.1 |
| 0 | 0 | 88.9 | ↓5.6 | 62.1 | ↓5.1 | 88.4 | ↓5.7 | 84.1 | ↓7.7 | 81.6 | ↓2.7 | 69.4 | ↓3.6 |

| | | **Regression Tasks** (lower is better) | | | | | | | | | | |
|---|---|---|---|---|---|---|---|---|---|---|---|---|
| M | K | FreeSolv | $\Delta$ | ESOL | $\Delta$ | Lipo | $\Delta$ | QM7 | $\Delta$ | QM8 | $\Delta$ | |
| 1 | 1 | **1.129** | - | **0.785** | - | 0.743 | - | **57.2** | - | **0.014** | - | |
| 0 | 1 | 1.313 | ↑0.184 | 0.834 | ↑0.049 | **0.708** | ↓0.035 | 64.6 | ↑7.4 | 0.016 | ↑0.002 | |
| 1 | 0 | 1.563 | ↑0.434 | 0.841 | ↑0.056 | 0.876 | ↑0.133 | 74.4 | ↑17.2 | 0.017 | ↑0.003 | |
| 0 | 0 | 1.944 | ↑0.815 | 0.978 | ↑0.193 | 0.845 | ↑0.102 | 77.9 | ↑20.7 | 0.017 | ↑0.003 | |

