# OpenReview forum: "Bi-level Contrastive Learning for Knowledge Enhanced Molecule Representations"
_ICLR.cc/2024/Conference — Submitted to ICLR 2024_

### Official Review · Reviewer_1pz2 · 2023-10-25

**Soundness:** 3 good
**Presentation:** 3 good
**Contribution:** 2 fair
**Rating:** 3
**Confidence:** 4

**Summary:**

This paper introduces Gode, a novel approach that integrates graph representations of individual molecules with multi-domain biomedical data from knowledge graphs. By pre-training two graph neural networks on different graph structures and employing contrastive learning, Gode effectively combines molecular structures with corresponding knowledge graph substructures. It achieves SOTA performance on 11 chemical property tasks.

**Strengths:**

The paper is well-written and easy to follow.

The authors conducted extensive experiments to investigate the proposed method under various settings and tasks.

**Weaknesses:**

The novelty of this paper may be limited as Gode’s modules, in my perspective, appear to originate from KANO. Gode reorganizes the knowledge graph (KG) and contrastive learning modules while incorporating a typical graph neural network (GNN) module. Additionally, the performance of Gode on most datasets is slightly better but comparable to that of KANO.

Regarding Figure 2, the ablation study results are interesting as even a minor adjustment in Gode can result in inferior performance compared to KANO.

Lastly, I believe the authors should conduct additional experiments to directly compare the proposed method with KANO. It appears that the authors may have overlooked this aspect, as many experiments in the current draft only compare alternative methods within Gode. For instance, enriching Figures 3-5 with the results of KANO would significantly enhance the persuasiveness of the findings.

**Questions:**

Please see Weaknesses.

---

> ### Author Response · Authors · 2023-11-18
> **Author Response to Reviewer 1pz2 - Part 1**
>
> Thank you for acknowledging the clarity, coherence, and comprehensive experimental scope of our paper. We address your concerns and answer your questions below. We also uploaded a revision and used blue to mark the new changes.
> ****
>
> ## **W1 (Novelty and Performance):**
>
> We appreciate your insight on the novelty of our work. For a comprehensive understanding, we recommend revisiting **our response to reviewer C94D - Parts 1 & 2**, where we elaborate on Gode's unique contributions and novelty.
>
> To clarify, KANO and Gode are very different frameworks where **KANO can only deal with Chemical Element KG** (which contains element-wise relationships) while **Gode can only deal with Molecule-Centric Biomedical KG** (which contains sophisticated biomedical relationships for molecules).**It's crucial to note that the types of KGs employed by these two frameworks are not interchangeable.**
>
> Achieving performance comparable to, or even surpassing, KANO on 10 out of 11 tasks - a benchmark method in our field - using a completely different knowledge graph should not be seen as a weakness. Instead, this accomplishment highlights the effectiveness of our framework.
>
> Below is a table delineating the key differences between the two KGs
>
> |                             |           Chemical Element KG (by KANO)            |                       MolKG (by Gode)                        |
> | :-------------------------- | :------------------------------------------------ | :---------------------------------------------------------- |
> | **Data Source**             |             Periodic Table of Elements             |       PubChemRDF (over 100 million molecules), PrimeKG       |
> | **Relationships (Triples)** |           chemical element-wise triples            |            extensive biomedical knowledge triples            |
> | **Entity Types**            |         2 types: Element, Functional Group         | 7 types: molecule, gene, disease, phenotype, drug, pathway, value |
> | **Scalability**             |  Hard to scale (as it is element-wise knowledge)   |          Highly scalable; Linear to the # molecules          |
> | **Data Update Frequency**   | Infrequent, as elemental data is relatively static | Regular updates, reflecting ongoing biomedical research advances |
>
> We are confident that the extensive data richness, superior scalability, and frequent updates inherent to the Molecule-Centric KG sub-graph from MolKG, along with its subsequent iterations, will establish it as an essential and highly effective modality for knowledge-enhanced molecular representation in the field.
> ****
>
> ## **W2 (Ablation Study Results in Figure 2):**
>
> We appreciate the opportunity to clarify the implications of our ablation study presented in Figure 2. It is crucial to emphasize that the variants we discuss in this figure are not minor adjustments but integral components of our framework. Each variant plays a significant role, and their removal can significantly impact performance, as we will detail:
>
> 1. **KGE (K-GNN embedding initialization):** This is a vital process in the pre-training of K-GNN, as further evidenced by Figure 4. Its exclusion leads to a substantial performance decrease, a finding consistent with the standard application of KGE in KANO. The impact of removing KGE is clearly demonstrated when comparing Case 1 and Case 3 in Figure 2.
> 2. **$\kappa$-hop:** The difference between a 3-hop and a 2-hop subgraph is significant, with the former having nearly 20 times more nodes and edges. This increase in complexity introduces noise, making the graph more challenging to learn from. This aspect is unique to our framework, as KANO's Chemical Element KG does not require exploration of this parameter. The performance variation between these configurations is evident when comparing Case 3 and Case 5 in Figure 2.
> 3. **Pretraining:** The self-supervised pretraining of K-GNN is crucial for aggregating local information within the molecule-centric KG subgraph into the central molecule node. The importance of this step is highlighted in Table 7 (Appendix G.1) of our revised paper.
> 4. **$\mathbf{h}_{\mathrm{f}}$:** The functional group feature, extracted using RDKit, is a commonly used attribute in property prediction, as seen in frameworks like GROVER and KANO. Omitting this feature, as shown by comparing Case 9 and Case 3, leads to a noticeable performance decline. This finding aligns with KANO's observations in their *Extended Data Fig. 1-a* in their paper [1].
>
> In summary, each component in our framework plays a substantial role in ensuring optimal performance. Our ablation study methodically examines these components, thereby solidifying the effectiveness of our approach.

---

> > ### Author Response · Authors · 2023-11-18
> > **Author Response to Reviewer 1pz2 - Part 2**
> >
> > ## **W3: Additional Comparison with KANO**
> >
> > Thank you for suggesting further experimental comparisons with KANO.
> >
> > **Regarding Figure 3:** KANO has already conducted a similar extensive experiment, detailed in *Extended Data Fig. 1-b* of their paper [1]. Due to the comprehensive nature of this experiment, replicating it within the limited timeframe of this rebuttal may not be feasible.
> >
> > **Regarding Figure 4:** The tasks illustrated in Figure 4 are specifically designed for the self-supervised pre-training of K-GNN within our Gode framework. These tasks are unique to our approach and differ significantly from the KANO framework, as elaborated in our response to W1. Therefore, a direct comparison in this context is not practical.
> >
> > **Regarding Figure 5:** In response to your feedback, we have included KANO's embeddings (enhanced by their Chemical Element KG) in our latest revised paper. This addition should provide a clearer comparison between Gode and KANO.
> >
> >
> >
> > ****
> >
> > **References**
> >
> > [1] Fang, Y., Zhang, Q., Zhang, N., Chen, Z., Zhuang, X., Shao, X., ... & Chen, H. (2023). Knowledge graph-enhanced molecular contrastive learning with functional prompt. *Nature Machine Intelligence*, 1-12.

---

> > > ### Comment · Reviewer_1pz2 · 2023-11-20
> > >
> > > Thank you for the valuable feedback. After carefully reviewing the comments from other reviewers and examining the KANO paper, I have decided to revise my score from 5 to 3. My reasons for this decision are as follows:
> > >
> > >
> > > 1. Deviation in experimental results. Reviewer ar3U highlighted significant deviations in the experimental results. It is not only for GROVER but also for KANO.  Most results of KANO reported in its paper are notably higher than the results obtained in your evaluation. For example, I observe discrepancies in the RMSE results across five benchmarks, as shown in Table 3:
> > >
> > >     | Methods  | FreeSolv | ESOL | Lipophilicity | QM7 | QM8 |
> > >     |---|---:|---:|---:|---:|---:|
> > >     | MolCLR (yours)  | 2.301 ± 0.247 | 1.113 ± 0.023 | 0.789 ± 0.009 | 90.0 ± 1.7 | 0.019 ± 0.013 |
> > >     | MolCLR (KANO paper)  | 2.301 ± 0.247 | 1.113 ± 0.023 | 0.789 ± 0.009 | 90.9 ± 1.7 | 0.0185 ± 0.013 |
> > >     | KANO (yours)  | 1.320 ± 0.244 | 0.902 ± 0.104 | 0.641 ± 0.012 | 66.5 ± 3.7 | 0.014 ± 0.001 |
> > >     | KANO (KANO paper)  | 1.142 ± 0.258 | 0.670 ± 0.019 | 0.566 ± 0.007 | 56.4 ± 2.8 | 0.0123 ± 0.000 |
> > >     | GODE  | 1.129 ± 0.314 | 0.785 ± 0.128 | 0.743 ± 0.043 | 57.2 ± 3.0 | 0.014 ± 0.001 |
> > >
> > >     It is unclear why only the performance of KANO is affected, while most results of MolCLR align with those reported in the KANO paper. One possibility is that KANO significantly outperforms the proposed GODE ... I also investigated the GitHub repository of KANO and found no reported issues regarding reproducibility.
> > >
> > > 2. The novelty. As also mentioned by Reviewers ar3U & C94D, this work is a combination of previous methods. Nevertheless, I don’t consider it a severe problem.
> > >
> > > 3. The concerns I raised in my initial review have not been adequately addressed.  Since the performance of the proposed GODE is not significantly better than KANO, it is crucial to conduct additional experiments comparing the proposed GODE with KANO, specifically in terms of training time and efficiency in leveraging KGs. Unfortunately, the authors have not addressed this issue.

---

> > > > ### Author Response · Authors · 2023-11-20
> > > > **Author Response to Reviewer 1pz2**
> > > >
> > > > Dear Reviewer,
> > > >
> > > > We would also be very grateful if you could suggest any other experiments you'd like us to undertake before the author-reviewer discussion deadline, alongside the thorough evaluation of KANO's pre-training that we are currently conducting.

---

> > > > ### Author Response · Authors · 2023-11-22
> > > > **Author Response to Reviewer 1pz2**
> > > >
> > > > Dear Reviewer,
> > > >
> > > > As the rebuttal deadline for our submission is rapidly approaching, we wanted to reach out to ensure that all your concerns have been addressed satisfactorily. Should there be any further questions or points of clarification needed, please let us know. We are ready and willing to provide additional information or engage in further discussion to address any issues.

---

> ### Author Response · Authors · 2023-11-20
> **Author Response to Reviewer 1pz2**
>
> Thank you for your valuable feedback.
>
> ****
>
> ### **Regarding the deviations in experimental results you highlighted:**
>
> **MolCLR:** The results of its default setting were sourced from the KANO paper, as detailed in Appendix D.2. We independently reproduced its pre-training using the GTransformer backbone and reported it as $\text{MolCLR}_{\text{GTrans}}$
>
> **GROVER:** The deviations you’ve observed are also documented in other studies, as well as in KANO's findings. We addressed this in our response to Reviewer ar3U (**W2&Q2**). The variance is evident in both scenarios: using the provided pre-trained models or after re-training them ourselves, which is corroborated by the discussions in their [issues repository](https://github.com/tencent-ailab/grover/issues).
>
> **KANO:** To our knowledge, we are the first to compare our work directly with KANO. This necessitated that we reproduce their pre-training process rather than rely on their available models. We adhered to their default setting, pre-training KANO over 50 epochs, which required approximately 21 hours on a single A6000 GPU.
>
> We assure you that our reported results are authentic and the outcome of a meticulous and repeatable process. Acknowledging the potential for variability due to non-deterministic factors (like how GROVER explained their reproducibility issue in their repo-readme) we took the additional step of running the pre-training process three times to confirm the stability of our findings.
>
> Should you wish, we invite you to reproduce the KANO pre-training process for further validation or to review additional papers that have attempted a similar reproduction. Considering the bias introduced by pre-training, ~~**we will also pre-train KANO as many times as possible before the rebuttal deadline, and report the standard deviation of the performance.**~~ Please see our results of extensive experiments for KANO below.
>
> We are confident in our results and remain open to discussing our methodology and findings in more detail.
>
> ****
>
> ### **For Additional Experiments**
>
> Thanks for reminding us again about this, but there are a few to note:
>
> In your original feedback, you mentioned you would like to see the direct comparison between KANO and Gode regarding experiments in Figures 3, 4, and 5.
>
> We responded that Figure 3 is not feasible to conduct during this rebuttal window, as KANO has already done this. We would need to pre-train KANO for $5 (\text{KG ratios}) \times 3 (\text{independent runs}) \times 21 (\text{hours/epoch} = 315 (\text{hours})$ to replicate this experiment.
>
> The experiment in Figure 4 is unique to Gode, as the tasks mentioned therein are defined by us. It is not applicable to KANO.
>
> For Figure 5, we have already included the embedding visualization of KANO.
>
> Regarding training time, KANO takes 21 hours and Gode takes 9 hours when $\kappa=2$, and 71 hours when $\kappa=3$. As we reported in Figure 2, the performance is better when $\kappa=2$. Thus, the pre-training of Gode is also more efficient than KANO.

---

> ### Author Response · Authors · 2023-11-21
> **Regarding  KANO's Performance**
>
> ## **Regarding  KANO's performance we reported**
>
> Dear Reviewer,
>
> In the past three days, we have **pre-trained 7 KANO models from scratch and tested each of them on downstream tasks.** Among these, KANO-1, KANO-2, and KANO-3 were trained on three A6000 GPUs, while the remaining four were trained on A100 GPUs separately.
>
> We have uploaded **all training logs for both pretraining and fine-tuning** here:
>
> **<[download link (training logs)](https://drive.google.com/file/d/1qTcUM5iEXivdeEDjSlcvKQpovURIW9Zf/view?usp=sharing)>**
>
> Additionally, we provide these 7 **pretrained KANO models** here:
>
> **<[download link (pretrained models)](https://drive.google.com/file/d/1roohZBpQTGdMbwMySGjXSto1nZKlIAeX/view?usp=sharing)>**
>
> We summarize the results below. We also discovered that the fine-tuning code provided by KANO's repository is not randomly seeded but is specifically set to 51, 52, and 53 by the author. Therefore, we tested each model with a randomly seeded setup. This is crucial to ensure consistency with the experimental setup described in their paper.
>
> **All experimental settings, except for the random seeds, are exactly the same as those in KANO's repo.**
>
> **Classification Tasks:**
>
> |        | BBBP           | SIDER          | ClinTox        | BACE           | Tox21          | ToxCast        |
> | ------ | -------------- | -------------- | -------------- | -------------- | -------------- | -------------- |
> | KANO-1 | 94.8 $\pm$ 3.6 | 61.8 $\pm$ 2.6 | 87.9 $\pm$ 2.1 | 87.4 $\pm$ 1.7 | 79.4 $\pm$ 0.7 | 69.8 $\pm$ 1.7 |
> | KANO-2 | 92.5 $\pm$ 1.7 | 63.2 $\pm$ 1.2 | 91.6 $\pm$ 5.2 | 91.9 $\pm$ 1.1 | 80.5 $\pm$ 0.8 | 71.0 $\pm$ 2.2 |
> | KANO-3 | 95.5 $\pm$ 2.1 | 62.2 $\pm$ 3.1 | 90.1 $\pm$ 0.4 | 87.7 $\pm$ 1.6 | 82.6 $\pm$ 1.2 | 71.4 $\pm$ 1.9 |
> | KANO-4 | 94.5 $\pm$ 1.7 | 63.0 $\pm$ 1.2 | 90.2 $\pm$ 3.0 | 87.5 $\pm$ 1.2 | 82.4 $\pm$ 0.6 | 71.7 $\pm$ 0.5 |
> | KANO-5 | 93.7 $\pm$ 1.8 | 64.3 $\pm$ 0.2 | 90.8 $\pm$ 4.7 | 82.5 $\pm$ 4.2 | 80.5 $\pm$ 0.5 | 69.2 $\pm$ 1.8 |
> | KANO-6 | 95.4 $\pm$ 1.2 | 62.7 $\pm$ 1.8 | 91.6 $\pm$ 0.6 | 86.6 $\pm$ 1.4 | 80.5 $\pm$ 1.9 | 70.1 $\pm$ 0.6 |
> | KANO-7 | 95.2 $\pm$ 3.0 | 61.8 $\pm$ 2.6 | 92.3 $\pm$ 0.6 | 88.7 $\pm$ 3.0 | 82.1 $\pm$ 1.2 | 70.8 $\pm$ 1.9 |
>
> **Regression Tasks:**
>
> |        | FreeSolv          | ESOL              | Lipophilicity     | QM7             | QM8               |
> | ------ | ----------------- | ----------------- | ----------------- | --------------- | ----------------- |
> | KANO-1 | 1.853 $\pm$ 0.250 | 0.827 $\pm$ 0.120 | 0.710 $\pm$ 0.038 | 61.4 $\pm$ 2.3  | 0.013 $\pm$ 0.002 |
> | KANO-2 | 1.567 $\pm$ 0.119 | 0.819 $\pm$ 0.104 | 0.698 $\pm$ 0.043 | 70.1 $\pm$ 16.4 | 0.017 $\pm$ 0.002 |
> | KANO-3 | 1.773 $\pm$ 0.200 | 0.824 $\pm$ 0.106 | 0.694 $\pm$ 0.032 | 61.4 $\pm$ 7.0  | 0.013 $\pm$ 0.003 |
> | KANO-4 | 1.836 $\pm$ 0.395 | 0.859 $\pm$ 0.070 | 0.635 $\pm$ 0.035 | 75.9 $\pm$ 3.4  | 0.013 $\pm$ 0.001 |
> | KANO-5 | 1.961 $\pm$ 0.546 | 0.798 $\pm$ 0.025 | 0.631 $\pm$ 0.012 | 66.9 $\pm$ 0.9  | 0.016 $\pm$ 0.001 |
> | KANO-6 | 1.871 $\pm$ 0.500 | 0.787 $\pm$ 0.118 | 0.636 $\pm$ 0.040 | 65.2 $\pm$ 7.4  | 0.014 $\pm$ 0.002 |
> | KANO-7 | 1.763 $\pm$ 0.480 | 0.710 $\pm$ 0.055 | 0.588 $\pm$ 0.017 | 63.8 $\pm$ 5.8  | 0.014 $\pm$ 0.002 |
>
> ~~We are still fine-tuning models on QM8, as it is time-consuming. We will also include it soon.~~ (done)
>
> We will update KANO's performance in Tables 2 and 3 based on those new results in our final revision.
>
>
> We sincerely hope that the **new results, along with the transparent training logs and models**, can address your concerns and doubts about KANO's performance reported by us.
>
> Please let us know if you have any other concerns.

---

> > ### Author Response · Authors · 2023-11-22
> > **Update to "Regarding KANO's Performance"**
> >
> > We strongly recommend the reviewers pre-train a KANO model by themselves if they still have concerns regarding the performance discrepancy.
> >
> > If they do not have time to do so, they can also download the pre-trained model provided by KANO's repo [1], and fine-tune the model on downstream tasks with random seeds, which will probably lead to similar results as follows:
> >
> > **<[our training logs for the provided pre-trained KANO by [1]](https://drive.google.com/file/d/1XCBhCNK3iPjG5gqXQwRcJ5l13DRWgiEE/view?usp=sharing)>**
> >
> > **Classification Tasks:**
> >
> > |                                          | BBBP           | SIDER          | ClinTox        | BACE           | Tox21          | ToxCast        |
> > | ---------------------------------------- | -------------- | -------------- | -------------- | -------------- | -------------- | -------------- |
> > | KANO (pre-trained model provided by [1]) | 94.5 $\pm$ 1.9 | 61.0 $\pm$ 0.6 | 92.6 $\pm$ 4.1 | 87.2 $\pm$ 4.2 | 81.7 $\pm$ 2.0 | 70.8 $\pm$ 0.7 |
> >
> > **Regression Tasks:**
> >
> > |                                          | FreeSolv         | ESOL              | Lipophilicity     | QM7            | QM8               |
> > | ---------------------------------------- | ---------------- | ----------------- | ----------------- | -------------- | ----------------- |
> > | KANO (pre-trained model provided by [1]) | 1.535$\pm$ 0.074 | 0.734 $\pm$ 0.101 | 0.635 $\pm$ 0.042 | 59.9 $\pm$ 1.7 | 0.014 $\pm$ 0.003 |
> >
> >
> >
> > ****
> >
> > **References**
> >
> > [1] KANO's official GitHub repo: https://github.com/HICAI-ZJU/KANO.

---

### Official Review · Reviewer_C94D · 2023-10-29

**Soundness:** 3 good
**Presentation:** 3 good
**Contribution:** 2 fair
**Rating:** 5
**Confidence:** 4

**Summary:**

In this work, the authors proposed GODE that integrates graph representations of individual molecules with multi-domain biochemical data from knowledge graphs. Graph neural network (GNN) was used to learn representation of molecular graphs and knowledge graphs. Contrastive learning was used to improve its predictive capabilities. Improved molecular property prediction results were demonstrated in experiments.

**Strengths:**

The clear and coherent structure help comprehension. The authors provide good insights into potential research directions.

**Weaknesses:**

The originality is light based on the following observations:
1.	The preliminary steps for pretraining the original molecule graph representation have previously been explored in the paper titled "Does GNN Pretraining Help Molecular Representation?" (Sun et al, 2022).
2.	Similarly, the pretraining strategies for bio Knowledge graph representation were also discussed in "Pre-training Graph Neural Networks for Molecular Representations: Retrospect and Prospect" (Xia et al, 2022).
3.	The construction of a Biomedical Knowledge Graph was prevously detailed in the paper "A unified drug–target interaction prediction framework based on knowledge graph and recommendation system" (Ye et al, 2021).
4.	Furthermore, the concept of contrastive learning between the molecular graph and an augmented knowledge graph for molecules was previously addressed in "Molecular Contrastive Learning with Chemical Element Knowledge Graph" (Fang et al, 2022).

This work is a combination of the previously mentioned research, with some customization. Additionally, =the performance of GODE appears to be close to that of "Molecular Contrastive Learning with Chemical Element Knowledge Graph" (Fang et al, 2022) and "Knowledge graph-enhanced molecular contrastive learning with functional prompt" (Fang et al, 2023).

**Questions:**

The author states the embedding produced by GODE is robust, but I do not see the corresponding discussion about robustness learning in the later sections. Could authors elaborate more?

Explain more about what the positive/negative classes are in contrast learning.

In page 4 "Embedding Initialization", four methods were cited. Which method was used in the experiments? If all of them were used, did the authors observe any differences in performance?

---

> ### Author Response · Authors · 2023-11-17
> **Author Response to Reviewer C94D - Part 1**
>
> Thanks for recognizing the comprehension and insightful research direction of our work. We address your concerns and answer your questions below. We also uploaded a revision and used blue to mark the new changes.
>
> ****
>
> **For Weaknesses:**
>
> 1. We acknowledge that molecule graph pre-training strategies have been well-studied, as in the paper by Sun et al. [1]. However, the focus of our work is to investigate how biomedical knowledge graphs can enhance molecule representations, rather than proposing new strategies for molecule graph pre-training.
>
> 2. The paper by Xia et al. [2] does not discuss pre-training strategies for bio-knowledge graphs. Instead, it mentions only the **Chemical Element KG**, as proposed by KCL [3], which is significantly different from our constructed biomedical knowledge graph, MolKG:
>
>    - **Chemical Element KG** is sourced from the Periodic Table of Elements and focuses on the attributes of chemical elements.
>    - **MolKG** is a molecule-centric biomedical KG, integrating molecule-tailored biomedical information from PubChemRDF (the largest KG for chemistry) and PrimeKG (a precision medicine-oriented KG).
>
> 3. Ye et al.'s work [4] does not construct its own biomedical knowledge graph but uses existing datasets (Luo's dataset, Hetionet, Yamanishi_08's dataset, and BioKG). None of these datasets contains the abundant molecules found in our downstream datasets. In contrast, our MolKG construction can scale to any molecules contained by PubChem or PrimeKG, with their relevant KG triples extracted. We detailed our construction steps in Appendix B, along with the processing scripts.
>
> 4. We would like to clarify some key differences in our approach compared to Fang et al, 2022 [3].
>
>    In both papers by KCL [3] and KANO [5], contrastive learning is conducted between **(1) the molecule graph** and **(2) the augmented molecule graph**, where (2) is obtained by linking element-wise KG triples from the Chemical Element KG directly to each element of the molecule.
>
>    In our work, differently, we conduct contrastive learning between **(1) the molecule graph**, and **(2) the central molecule node in (1)'s corresponding KG sub-graph**. Unlike KCL or KANO, we do not augment molecule graphs explicitly but treat the molecule as a node in the KG sub-graph. We also designed a self-supervised pre-training strategy (Eq. 3) for this KG sub-graph, the effectiveness of which is demonstrated in Appendix G.1.
>
>    Please also feel free to read Appendix F for more details regarding our comparison to similar works.
>
> 5. As explained in points 1 to 4, our work is not a simple combination of the mentioned research but introduces major innovations:
>
>    1. A tailored molecule-centric KG providing comprehensive biomedical knowledge.
>    2. A novel integration of molecular graphs with biomedical domain knowledge, treating molecules as nodes in KG sub-graphs.
>
>    KANO [5] is an improved version of KCL [3] in both simplicity and performance stability. Thus, our comparison with KANO, as shown in Tables 2 and 3, underscores our unique contribution.

---

> > ### Author Response · Authors · 2023-11-17
> > **Author Response to Reviewer C94D - Part 2**
> >
> > We believe that **being comparable to (and consistently outperforming) KANO means our method is good**, for the following reasons:
> >
> > - **Distinct Information Utilization**: We use a completely different information source (molecule-centric biomedical KG), unlike KANO's Chemical Element KG, to enhance molecule representations and improve molecular property prediction.
> >
> > - **Complementary to KANO's Contribution**: The molecule graph can be augmented either explicitly through KANO's element-wise linking or implicitly through our Gode's graph-as-a-node approach, or by exploring the combination of both.
> >
> > - **Scalability and Performance Potential**: Figure 3 in our paper suggests Gode's performance may increase with a larger MolKG. Currently, we have only extracted triples for 65,454 molecules from PubChemRDF, which contains over 100 million molecules, indicating significant potential for further performance improvements.
> >
> > - **Performance Improvement of Past SOTA Models:** We list the performance improvement over the previous SOTA reported by the past papers (including ours) on their common classification tasks:
> >
> >   |                 |           BBBP           |         SIDER         |       ClinTox       |           BACE           |          Tox21          |      | Avergae |
> >   | :-------------- | :----------------------: | :-------------------: | :-----------------: | :----------------------: | :---------------------: | :--: | :-----: |
> >   | **MolCLR**  [6] |   -19.2% (over N-GRAM)   |    -0.1% (over RF)    | +3.0% (over D-MPNN) |     +2.7% (over RF)      |    -2.4% (over SVM)     |      |  -3.3%  |
> >   | **GEM** [7]     | +2.0% (over PretrainGNN) |  +0.6% (over N-GRAM)  | -0.6% (over D-MPNN) | +1.3% (over PretrainGNN) | 0.0% (over PretrainGNN) |      |  +0.7%  |
> >   | **KANO** [5]    |    +3.6% (over CMPNN)    | +2.0% (over GraphMVP) | +4.2% (over DMPNN)  |     +5.9% (over GEM)     |    +3.6% (over MPNN)    |      |  +3.9%  |
> >   | **Gode** (ours) |    +0.9% (over KANO)     |   +5.3% (over KANO)   |  +0.5% (over KANO)  |    +1.5% (over KANO)     |   +2.1% (over DMPNN)    |      |  +2.1%  |
> >
> >
> >
> > ****
> >
> > **References**
> >
> > [1] Sun, R., Dai, H. and Yu, A.W., 2022. Does GNN Pretraining Help Molecular Representation? *NeurIPS 2022*.
> >
> > [2] Xia, J., Zhu, Y., Du, Y. and Li, S.Z., 2022, June. Pre-training graph neural networks for molecular representations: retrospect and prospect. In *ICML 2022 2nd AI for Science Workshop*.
> >
> > [3] Fang, Y., Zhang, Q., and others. Molecular Contrastive Learning with Chemical Element Knowledge Graph. *AAAI 2022*.
> >
> > [4] Ye, Q., Hsieh, C.Y., Yang, Z., and others. 2021. A unified drug–target interaction prediction framework based on knowledge graph and recommendation system. *Nature communications*.
> >
> > [5] Fang, Y., Zhang, Q., and others. (2023). Knowledge graph-enhanced molecular contrastive learning with functional prompt. *Nature Machine Intelligence*, 1-12.
> >
> > [6] Wang, Y., Wang, J., Cao, Z. and Barati Farimani, A., 2022. Molecular contrastive learning of representations via graph neural networks. *Nature Machine Intelligence*, *4*(3), pp.279-287.
> >
> > [7] Fang, X., Liu, L., Lei, J., He, D., Zhang, S., Zhou, J., Wang, F., Wu, H. and Wang, H., 2022. Geometry-enhanced molecular representation learning for property prediction. *Nature Machine Intelligence*, *4*(2), pp.127-134.

---

> > > ### Author Response · Authors · 2023-11-18
> > > **Author Response to Reviewer C94D - Part 3**
> > >
> > > **For Your Questions:**
> > >
> > > **Q1 (Robustness of Gode):** We believe that the leading performance of Gode across 10/11 datasets (which include 676 tasks in total) can indicate the robustness of the learned molecule representation.
> > >
> > > **Q2 (Positive/Negative Samples in Contrastive Learning):**
> > >
> > > Each positive sample in $\mathcal{D}^{+} =$ {$(m_i, G\_{\mathrm{sub}(m\_i, \kappa)}), y\_i = 1$}$\_{N\_p}$  is composed by **a molecule graph $m\_i$** and **its corresponding $\kappa$-hop molecule-centric KG sub-graph $G\_{\mathrm{sub}(m\_i, \kappa)}$** extracted from MolKG.
> > >
> > > Each negative sample in $\mathcal{D}^{-} =$ {$(m\_i, G\_{\mathrm{sub}(m\_j, \kappa)})\_{j \neq i}, y\_i = 0$}$\_{N-N\_p}$ is composed by **a molecule graph $m\_i$** and **a non-corresponding $\kappa$-hop molecule-centric KG sub-graph $G_{\mathrm{sub}(m_j, \kappa)})_{j \neq i}$** sampled from MolKG. To make the task more challenging, we further divide $\mathcal{D}^{-}$ into $\mathcal{D}^{-}\_{rand}$, and $\mathcal{D}^{-}\_{nbr}$, which are (1) randomly sampled from all negative molecule-centric KG sub-graphs, and (2) sampled from the sub-graphs of the neighbor molecule nodes connected to the positive molecule node ($m\_i$), respectively.
> > >
> > > We also improved the corresponding content in the revised paper for clarity.
> > >
> > > **Q3 (Embedding Initialization):** We use TransE [8] for the KG embedding (KGE) initialization, as we specified in ***Section 4.1 - Implementation*** and ***Appendix D.1***. For KGE methods, we tried TransE, RotatE [9], DistMult [10], and TuckER [11], among which TransE performed the best regarding the final performance on the property prediction tasks.
> > >
> > >
> > > ****
> > >
> > > **References**
> > >
> > > [8] Bordes, A., Usunier, N., Garcia-Duran, A., Weston, J. and Yakhnenko, O. Translating embeddings for modeling multi-relational data. NIPS 2013.
> > >
> > > [9] Sun, Z., Deng, Z.H., Nie, J.Y. and Tang, J.. Rotate: Knowledge graph embedding by relational rotation in complex space. ICLR 2019.
> > >
> > > [10] Yang, B., Yih, W.T., He, X., Gao, J. and Deng, L. Embedding entities and relations for learning and inference in knowledge bases. ICLR 2015.
> > >
> > > [11] Balažević, I., Allen, C. and Hospedales, T.M.. Tucker: Tensor factorization for knowledge graph completion. EMNLP-IJCNLP 2019.

---

> ### Comment · Reviewer_C94D · 2023-11-22
> **Effects of KG and CL**
>
> Please remind me if ablation experiments (or sth alike) were conducted to reveal the combinatorial effects of (a) using a different KG, and (b) using a different CL method.

---

> > ### Comment · Reviewer_C94D · 2023-11-22
> > **Thanks for providing all the above addition information.**
> >
> > Will consider them in deciding my final score.

---

> ### Author Response · Authors · 2023-11-22
> **Thanks for your questions**
>
> Dear Reviewer,
>
> Thank you for your questions. We are pleased to provide prompt responses as follows:
>
> **(a) Different KGs?**
>
> Yes, we have explored this aspect:
>
> - **Figure 3**: Shows the results of applying three different KGs in varying ratios (20%, 50%, 80%) to the original MolKG. KANO also did such a study in their "Extended Data Fig. 1".
> - **Author Response to Reviewer afSU - Part 2 - Q1**: We present new results with four different KGs, each with unique relation types. We will include a more comprehensive version of this study in our final revision.
>
> Regarding testing on KGs other than MolKG, the answer is no. Our method is specifically designed for molecule-centric KGs, such as MolKG, which is currently the only KG of such type. Similarly, the KANO framework is tailored for the Chemical Element KG they proposed. The KG types are not interchangeable between different frameworks like Gode and KANO. KANO also only studies the Chemical Element KG.
>
> **(b) Different CL Method?**
>
> No, we did not perform an ablation study on different contrastive learning methods. This decision aligns with the standard practices in the field, as evidenced by similar methodologies in notable works like MolCLR and KANO. These studies, too, did not explore alternative contrastive learning methods. Nevertheless, we would love to conduct this study and hopefully integrate it into our final revision.
>
> We hope these clarifications enhance your understanding of our proposed method.

---

### Official Review · Reviewer_ar3U · 2023-10-30

**Soundness:** 2 fair
**Presentation:** 3 good
**Contribution:** 2 fair
**Rating:** 3
**Confidence:** 4

**Summary:**

In this paper, the authors propose a molecular representation learning method, GODE, based on knowledge graphs. Specifically, GODE trains two graph neural networks, one for the molecule and another for its corresponding knowledge graph subgraph encoders. The molecule encoder employs the pretraining approach inspired by GROVER, while the knowledge subgraph encoder follows a GINE-like strategy. After pretraining, both are further trained through contrastive learning, and the learned representations are subsequently applied to downstream tasks.

**Strengths:**

1. The motivation of the paper is clear and reasonable. This paper considers applying knowledge graphs to molecular representation learning, which differs from the majority of methods that rely solely on information within molecules. This strategy allows for the incorporation of more domain knowledge, potentially leading to better representations.

2. The paper is well-structured, and the steps of the method are easy to follow, enhancing the accessibility and understanding of the proposed approach.

**Weaknesses:**

1. My primary concern stems from the experimental setup's consistency. The baseline methods used for comparison seem to follow different settings, as the authors mention in the appendix that they have reproduced these baselines using their official implementations. Some of these baselines follow the settings from [1], while others have their own network structures. To validate the effectiveness of their representation learning, experiments should ideally be conducted under the same settings, including the same backbone model, node/edge feature selection, and other relevant parameters. Different representation learning methods can yield different results when applied to different settings. For instance, the outcome of using Grover on GIN might differ from using it on GTransformer[2]. If each method is used with different settings, it becomes challenging to assess the effectiveness of the proposed method. The final results may be influenced by the quality of the chosen backbone models, making it difficult to draw meaningful comparisons and conclusions about the proposed method's performance. Therefore, it is essential to standardize the experimental settings to ensure a fair and consistent evaluation of all methods.

2. Furthermore, the experimental results for GROVER appear to exhibit significant deviations from what is reported in the original paper. This raises concerns about whether the authors may have overlooked or omitted some critical details during the reproduction process. It is imperative for the authors to thoroughly investigate and address any potential discrepancies in the results, ensuring that the experiments align closely with the findings presented in the GROVER paper.

3. This paper introduces a highly promising direction for using knowledge graphs in representation learning. However, the method section appears to be overcomplicated, and its technical contributions seem rather limited. Both the molecular and knowledge graph encoders undergo separate pretraining, followed by contrastive learning, which makes the pretraining process redundant. It is recommended that the authors poform more ablation studies on whether all these steps are neccessary.

[1] Hu W, Liu B, Gomes J, et al. Strategies for pre-training graph neural networks[J]. arXiv preprint arXiv:1905.12265, 2019.

[2] Zhang, Zaixi, et al. "Motif-based graph self-supervised learning for molecular property prediction." Advances in Neural Information Processing Systems 34 (2021): 15870-15882.

**Questions:**

1. See the major concern above.
2. Why is the GROVER results in the paper signigicantly lower than the reported results in the original paper?

---

> ### Author Response · Authors · 2023-11-16
> **Author Response to Reviewer ar3U**
>
> Thank you for appreciating our paper's clear motivation, innovative use of knowledge graphs in molecular representation, and the accessible structure of our method. We address your concerns and answer your questions below. We also uploaded a revision and used blue to mark the new changes.
>
> ****
>
> **W1: Experiments should ideally be conducted under the same settings.**
>
> Thank you for highlighting the importance of uniform experimental settings across different models. In our study, we paid careful attention to this aspect to ensure a fair and meaningful comparison of our GODE method with the baseline models.
>
> To address your concern regarding the standardization of experimental settings, we acknowledge that our original description in Appendix D may have caused some confusion. To clarify, this meant that while we utilized the original model architectures as provided by the authors, we also meticulously standardized all other experimental settings for a fair comparison. Our goal was to ensure that each method was evaluated on a consistent and comparable basis.
>
> Taking your feedback into consideration, we have further refined our experimental design for an even more equitable comparison. Specifically, we have now reproduced $\text{MolCLR}\_{\text{GTrans}}$ using the GTransformer backbone and $\text{KANO}\_\text{CMPNN}$ with its default CMPNN backbone as specified in the KANO paper. It's crucial to note that the results for GROVER and KANO presented in our initial submission were both based on the GTransformer backbone.
>
> In our revised submission, we have added a new section, ***Appendix D.2 Baseline Models***, which provides detailed insights into the experimental settings used for each baseline model. This addition aims to offer a transparent and comprehensive view on how we ensured uniformity in our comparative analysis.
>
> The results reflecting these changes also have been updated in ***Table 2*** and ***Table 3*** of our revised paper (in blue for easy identification). We believe these enhancements and clarifications in our methodology significantly strengthen the validity of our findings and address your concerns regarding the experimental setup's consistency.
>
>
>
> **W2&Q2:  The experimental results for GROVER appear to exhibit significant deviations from what is reported in the original paper.**
>
> We appreciate your concern regarding the observed deviations in GROVER's performance in our study compared to the results reported in its original paper [1]. We would like to emphasize that such discrepancies are not unique to our research. Similar deviations have been reported in other significant studies, including MGSSL [2], GEM [3], Mole-bert [4], and KANO [5], all of which were published in highly respected venues.
>
> Our experimental settings strictly adhered to the guidelines and configurations outlined in the original GROVER paper.
>
>
>
> **W3: Recommendation for Additional Ablation Studies**
>
> Thank you for your insightful recommendation to further investigate the necessity of the various steps in our methodology. In response, we have conducted detailed ablation studies, the results of which are presented in ***Appendix G.1 - Table 7*** of our revised manuscript.
>
> These studies decisively highlight the integral roles of both M-GNN and K-GNN pre-training within our framework. Each pre-training phase is shown to uniquely and significantly contribute to the overall enhancement of our model's performance. This evidence robustly counters the notion of redundancy in our methodological approach.
>
> We hope that these findings will satisfactorily address your concerns about the steps previously perceived as redundant in our methodology.
>
>
> ****
> **References**
>
> [1] Rong, Y., Bian, Y., Xu, T., Xie, W., Wei, Y., Huang, W. and Huang, J.. Self-supervised graph transformer on large-scale molecular data. In *NeurIPS* 2020.
>
> [2] Zhang, Z., Liu, Q., Wang, H., Lu, C. and Lee, C.K. Motif-based graph self-supervised learning for molecular property prediction. In *NeurIPS* 2021.
>
> [3] Fang, X., Liu, L., Lei, J., He, D., Zhang, S., Zhou, J., Wang, F., Wu, H. and Wang, H., 2022. Geometry-enhanced molecular representation learning for property prediction. *Nature Machine Intelligence*, *4*(2), pp.127-134.
>
> [4] Xia, J., Zhao, C., Hu, B., Gao, Z., Tan, C., Liu, Y., Li, S. and Li, S.Z. Mole-bert: Rethinking pre-training graph neural networks for molecules. In *ICLR* 2023.
>
> [5] Fang, Y., Zhang, Q., Zhang, N., Chen, Z., Zhuang, X., Shao, X., ... & Chen, H. (2023). Knowledge graph-enhanced molecular contrastive learning with functional prompt. *Nature Machine Intelligence*, 1-12.

---

> > ### Comment · Reviewer_ar3U · 2023-11-21
> >
> > I appreciate the authors' efforts in revising their manuscript to address several of my concerns. However, I still have some doubts regarding Q2, as GROVER has provided fine-tuned models for reproduction in their repository (https://github.com/tencent-ailab/grover).
> >
> > Additionally, I am thankful that the authors provide an additional study regarding each part in their method. However, I still have concerns related to the components in the proposed method. While the experiments only empirically evaluate the performance of each embedding model, it is still not clear why the authors select these three models as the backbone. Are there theoretical insights?  In addition, since different embeddings are derived by different encoders, this mechanism resembles ensemble models. It is worth considering that such an approach may come with a higher runtime overhead compared to previous baselines that use only one model to generate an embedding for prediction. While the method has achieved some improvement, it is important to take into account the runtime complexity and potential practical implications of this approach.

---

> > > ### Author Response · Authors · 2023-11-22
> > > **Author's Response to Reviewer ar3U**
> > >
> > > ## **Regarding GROVER's performance**
> > >
> > > Thank you for your valuable feedback regarding GROVER's performance. In response, we have incorporated the fine-tuned models as suggested, and our findings are presented in the tables below. We also include the results of other studies' reproduction of GROVER, highlighting the issue of reproducibility.
> > >
> > > **Performance on Classification Tasks:**
> > >
> > > |                                                           | BBBP           | SIDER          | ClinTox         | BACE           | Tox21          | ToxCast        |
> > > | --------------------------------------------------------- | -------------- | -------------- | --------------- | -------------- | -------------- | -------------- |
> > > | Fine-tuned $\text{GROVER}_{\text{Large}}$ provided by [1] | 93.9 $\pm$ 1.9 | 65.8 $\pm$ 2.3 | 94.4 $\pm$ 2.1  | 89.4 $\pm$ 2.9 | 83.1 $\pm$ 2.5 | 73.7 $\pm$ 1.0 |
> > > | GROVER reproduced by KANO [2]                             | 86.8 $\pm$ 2.2 | 61.2 $\pm$ 2.5 | 70.3 $\pm$ 13.7 | 82.4 $\pm$ 3.6 | 80.3 $\pm$ 2.0 | 56.8 $\pm$ 3.4 |
> > > | GROVER reproduced by GEM [3]                              | 69.5 $\pm$ 0.1 | 65.4 $\pm$ 0.1 | 76.2 $\pm$ 3.7  | 81.0 $\pm$ 1.4 | 73.5 $\pm$ 0.1 | 65.3 $\pm$ 0.5 |
> > > | GROVER reproduced by Mole-BERT [4]                        | 66.9 $\pm$ 2.1 | 61.0 $\pm$ 1.0 | 77.7 $\pm$ 2.7  | 73.0 $\pm$ 3.3 | 73.6 $\pm$ 0.7 | 62.3 $\pm$ 0.6 |
> > > | GROVER reproduced by us                                   | 86.2 $\pm$ 3.9 | 57.6 $\pm$ 1.6 | 74.7 $\pm$ 4.4  | 82.5 $\pm$ 4.4 | 76.9 $\pm$ 2.3 | 66.7 $\pm$ 2.6 |
> > >
> > > **Performance on Regression Tasks:**
> > >
> > > |                                                           | FreeSolv          | ESOL              | Lipophilicity     | QM7            | QM8               |
> > > | --------------------------------------------------------- | ----------------- | ----------------- | ----------------- | -------------- | ----------------- |
> > > | Fine-tuned $\text{GROVER}_{\text{Large}}$ provided by [1] | 1.544 $\pm$ 0.397 | 0.831 $\pm$ 0.120 | 0.560 $\pm$ 0.036 | 72.7 $\pm$ 3.8 | 0.013 $\pm$ 0.002 |
> > > | GROVER reproduced by KANO [2]                             | 1.947 $\pm$ 0.615 | 1.423 $\pm$ 0.288 | 0.823 $\pm$ 0.010 | 91.3 $\pm$ 1.9 | 0.018 $\pm$ 0.001 |
> > > | GROVER reproduced by GEM [3]                              | 2.272 $\pm$ 0.051 | 0.895 $\pm$ 0.017 | 0.823 $\pm$ 0.010 | 92.0 $\pm$ 0.9 | 0.022 $\pm$ 0.000 |
> > > | GROVER reproduced by Mole-BERT [4]                        | -                 | -                 | -                 | -              | -                 |
> > > | GROVER reproduced by us                                   | 2.445 $\pm$ 0.761 | 1.028 $\pm$ 0.145 | 0.890 $\pm$ 0.050 | 95.3 $\pm$ 5.6 | 0.020 $\pm$ 0.003 |
> > >
> > >
> > >
> > > ****
> > >
> > > **References**
> > >
> > > [1] GROVER's Official Github Repo: https://github.com/tencent-ailab/grover
> > >
> > > [2] Fang, Y., Zhang, Q., Zhang, N., Chen, Z., Zhuang, X., Shao, X., ... & Chen, H. (2023). Knowledge graph-enhanced molecular contrastive learning with functional prompt. *Nature Machine Intelligence*, 1-12.
> > >
> > > [3] Fang, X., Liu, L., Lei, J., He, D., Zhang, S., Zhou, J., Wang, F., Wu, H. and Wang, H., 2022. Geometry-enhanced molecular representation learning for property prediction. *Nature Machine Intelligence*, *4*(2), pp.127-134.
> > >
> > > [4] Xia, J., Zhao, C., Hu, B., Gao, Z., Tan, C., Liu, Y., Li, S. and Li, S.Z. Mole-bert: Rethinking pre-training graph neural networks for molecules. In *ICLR* 2023

---

> ### Author Response · Authors · 2023-11-22
> **Author's Response to Reviewer ar3U**
>
> We are happy to know that several of your concerns have been addressed by us. We answer your new questions and address your new concerns below.
>
> **(1) "GROVER has provided fine-tuned models"**
>
> As previous works (e.g., GEM, KANO) fine-tuned GROVER by themselves and reported the performance, we keep consistent with such a setting. We believe It is not rigorous to use the provided fine-tuned models as there is a possibility that they have been trained with the test data. Furthermore, GROVER provides an efficient framework to fine-tune the pre-trained model on downstream tasks. It only takes a few minutes to fine-tune a model on a task.
>
> Nevertheless, we will provide you with the results of those provided fine-tuned models soon, ~~**in our later response** to you.~~ **Please see the results below.**
>
> **(2) "Why select GTransformer, MPNN, and GIN as the backbones to test?"**
>
> We believe GTransformer is a sophisticated backbone to engage both edge and node information mutually, whereas MPNN and GIN are simpler but show good performance in Tables 2 and 3. Note that most previous works like KANO did not study the backbone, and took the best-performing GNN (e.g., CMPNN for KANO) as their backbone.
>
> **(3) "High runtime overhead due to different encoders"**
>
> We mentioned _"During fine-tuning, embeddings from K-GNN remain fixed, updating only the parameters of M-GNN"_ in **Section 4.1 - Implementation**, which means the embedding from K-GNN is **pre-computed and simply concatenated** to the M-GNN embedding as an additional feature (together with the extracted functional group feature) during fine-tuning or prediction. This means the fine-tuning/prediction is almost as efficient as the fine-tuning/prediction with M-GNN (e.g., GTransformer) itself.
>
> Here is an example for your information: fine-tuning one epoch on the BACE task takes 10.2 seconds for Gode while taking 9.6 seconds for GROVER (GTransformer).

---

> ### Author Response · Authors · 2023-11-22
> **Author's Response to Reviewer ar3U**
>
> Dear Reviewer,
>
> As the rebuttal deadline for our submission is rapidly approaching, we wanted to reach out to ensure that all your concerns have been addressed satisfactorily. Should there be any further questions or points of clarification needed, please let us know. We are ready and willing to provide additional information or engage in further discussion to address any issues.

---

> ### Author Response · Authors · 2023-11-22
> **Additional 10-fold experiments of GROVER**
>
> To thoroughly validate the results obtained from GROVER, we undertook an additional fine-tuning of the pre-trained GROVER model (specifically the large version available in their repository [1]) using a 10-fold cross-validation approach, as opposed to the 3-fold setting originally described in the paper. The detailed results of this rigorous re-evaluation are presented below.
>
> For complete transparency and to facilitate further analysis, we have uploaded both the short and long versions of the training logs. These can be accessed via the following link:
>
> **<[10-fold GROVER fine-tuning logs](https://drive.google.com/file/d/1jMs82xI6DVxsRa1v_DTUQfhXDLMxI6ZI/view?usp=sharing)>**
>
> **Classification Tasks:**
>
> |                                | BBBP           | SIDER          | ClinTox        | BACE           | Tox21          | ToxCast        |
> | ------------------------------ | -------------- | -------------- | -------------- | -------------- | -------------- | -------------- |
> | GROVER (10-fold averaged)      | 86.8 $\pm$ 2.9 | 56.1 $\pm$ 3.0 | 74.8 $\pm$ 6.3 | 80.1 $\pm$ 4.8 | 76.9 $\pm$ 3.1 | 65.6 $\pm$ 2.4 |
> | Best within 10 folds           | 91.6           | 62.1           | 83.5           | 86.6           | 81.7           | 70.2           |
> | Worst within 10 folds          | 81.3           | 51.3           | 63.2           | 71.0           | 71.7           | 61.3           |
> | GROVER (reported in our paper) | 86.2 $\pm$ 3.9 | 57.6 $\pm$ 1.6 | 74.7 $\pm$ 4.4 | 82.5 $\pm$ 4.4 | 76.9 $\pm$ 2.3 | 66.7 $\pm$ 2.6 |
>
> **Regression Tasks:**
>
> |                                | FreeSolv          | ESOL              | Lipophilicity     | QM7             | QM8               |
> | ------------------------------ | ----------------- | ----------------- | ----------------- | --------------- | ----------------- |
> | GROVER (10-fold averaged)      | 2.422 $\pm$ 0.934 | 1.005 $\pm$ 0.185 | 0.910 $\pm$ 0.052 | 95.7 $\pm$ 10.4 | 0.020 $\pm$ 0.003 |
> | Best within 10 folds           | 1.120             | 0.803             | 0.801             | 78.4            | 0.016             |
> | Worst within 10 folds          | 3.294             | 1.480             | 0.961             | 104.9           | 0.024             |
> | GROVER (reported in our paper) | 2.445 $\pm$ 0.761 | 1.028 $\pm$ 0.145 | 0.890 $\pm$ 0.050 | 95.3 $\pm$ 5.6  | 0.020 $\pm$ 0.003 |
>
> The results from the 10-fold validation align closely with the 3-fold results we reported in the paper, thereby reinforcing the validity of the GROVER performance that we initially presented.
>
>
>
> ****
>
> **References**
>
> [1] https://github.com/tencent-ailab/grover

---

> > ### Comment · Reviewer_ar3U · 2023-11-22
> >
> > Thank the authors for conducting these experiments. I will consider these after discussing with other reviewers.

---

> > > ### Author Response · Authors · 2023-11-22
> > > **Thank you**
> > >
> > > Thank you! Should you have any other major concerns, please let us know. We are open to further feedback and eager to address any issues you might have.

---

### Official Review · Reviewer_afSU · 2023-11-08

**Soundness:** 3 good
**Presentation:** 4 excellent
**Contribution:** 3 good
**Rating:** 8
**Confidence:** 4

**Summary:**

The paper proposes a molecule representation learning method, GODE,  that incorporates information from the molecular graph structure as well as domain specific biochemical information from knowledge graphs. The method leverages two pre-trained graph neural networks (GNN); one trained on the molecular graph and another trained on the relevant biochemical knowledge (sub)graphs. The GNNs are trained using contrastive learning to fuse the two complementary graph information. Downstream finetuning experiments suggest that the proposed method exhibits competitive performance in 11 chemical property prediction tasks compared to other molecule property pre-training methods.

**Strengths:**

*Proposed method shows strong performance in 11 molecule property prediction tasks from moleculenet

*Some nice comprehensive ablation studies, eg figure 3 and performance across different knowledge graph sizes

*Paper is well structured and well written

**Weaknesses:**

*Number of seeds (3) in experiments seems pretty low

*Missing some of the larger benchmark datasets. (eg HIV, PCBA). Any explanations why? (I’m not suggesting those are particularly great benchmark datasets)

**Questions:**

*Amount of relevant information in a knowledge graph likely varies significantly for a particular molecule. Eg aspirin is very well studied compared to other molecules. Any thoughts on when the knowledge graph information is useful or even harmful for a particular molecule property prediction task?

*In Table 2 and Table 3: would be clearer if there is some visual grouping of the methods that leverage molecule information only vs methods that additionally leverage knowledge graph information

*Figure 3: any thoughts on what affects the steepness of the improvement as knowledge graph size is increased? Eg size of the fine tuning dataset, chemical similarity of the fine tuning dataset compared to the knowledge graph, etc?

*Why wasn’t the Ye et al, 2021 model benchmarked?

---

> ### Author Response · Authors · 2023-11-16
> **Author Response to Reviewer afSU - Part 1**
>
> Thank you for your insightful and constructive feedback on our submission. We greatly appreciate your recognition of the strengths of our work. We address your concerns and answer your questions below. We also uploaded a revision and used blue to mark the new changes.
>
> ****
>
> **W1: Number of seeds in the experiment**
>
> Thanks for pointing out that the number of seeds (3) is low in experiments. While we recognize the value in your suggestion that using a greater number of seeds could yield more robust and convincing results, this experimental setup was chosen to maintain consistency with previous works [1, 2, 3], as suggested by MoleculeNet's paper [4]. We value your suggestion and will explore the use of additional seeds in our future research.
>
>
>
> **W2: Missing some of the larger benchmark datasets. (e.g., HIV, PCBA).**
>
> We appreciate your observation regarding the omission of larger datasets like HIV and PCBA in our study. Our primary focus was to explore how biochemical knowledge graphs can enhance molecule representation for property prediction. To maintain a consistent and comparable framework, we chose to validate our method on datasets used by our backbone - GROVER [1], a popular model that leverages molecule information only.
>
> We acknowledge the potential value of extending our approach to larger datasets, as this could further validate and strengthen our findings. However, constructing a more extensive MolKG that includes molecules from these datasets, along with conducting K-GNN pre-training and contrastive learning with the new KG, requires substantial time beyond this author's response window. Thus, we decided to include those results in our final revision.
>
>
>
> ***
>
> **References**
>
> [1] Rong, Y., Bian, Y., Xu, T., Xie, W., Wei, Y., Huang, W. and Huang, J.. Self-supervised graph transformer on large-scale molecular data. In *NeurIPS* 2020.
>
> [2] Zhang, Z., Liu, Q., Wang, H., Lu, C. and Lee, C.K. Motif-based graph self-supervised learning for molecular property prediction. In *NeurIPS* 2021.
>
> [3] Fang, Y., Zhang, Q., Zhang, N., Chen, Z., Zhuang, X., Shao, X., ... & Chen, H. (2023). Knowledge graph-enhanced molecular contrastive learning with functional prompt. *Nature Machine Intelligence*, 1-12.
>
> [4] Wu, Z., Ramsundar, B., Feinberg, E.N., Gomes, J., Geniesse, C., Pappu, A.S., Leswing, K. and Pande, V., 2018. MoleculeNet: a benchmark for molecular machine learning. *Chemical science*, *9*(2), pp.513-530.

---

> ### Author Response · Authors · 2023-11-16
> **Author Response to Reviewer afSU - Part 2**
>
> **Q1: When the knowledge graph information is useful or even harmful for a particular molecule property prediction task?**
>
> This is an insightful question. In our work using GINE for K-GNN, both node (entity) and edge (relation) information in the knowledge graph can influence molecular representation. We recognize the importance of identifying which information in the KG is beneficial or detrimental and plan to conduct a systematic study on this in the future. To this end, we conducted a preliminary experiment to understand the impact of different relations on molecular property prediction tasks:
>
> | Knowledge Graph                                              | BBBP                    | BACE                    |
> | ------------------------------------------------------------ | ----------------------- | ----------------------- |
> | MolKG                                                        | 94.5                    | 91.8                    |
> | w/o relation "indication"                                    | 93.8 ($\downarrow$ 0.7) | 91.6 ($\downarrow$ 0.2) |
> | w/o relations "xlogp3" and "xlogp3-aa"                       | 93.5 ($\downarrow$ 1.0) | 91.1 ($\downarrow$ 0.7) |
> | w/o relations "tautomer_count" and "covalent_unit_count"     | 94.3 ($\downarrow$ 0.2) | 90.9 ($\downarrow$ 0.9) |
> | w/o relations "neighbor_2d", "neighbor_3d", and "has_same_connectiviy" | 94.9 ($\uparrow$ 0.4)   | 91.3 ($\downarrow$ 0.5) |
>
> For the BBBP task, removing "xlogp3" and "xlogp3-aa" led to a significant decrease in performance, suggesting their vital role in the model. In contrast, omitting "neighbor_2d", "neighbor_3d", and "has_same_connectivity" resulted in a slight improvement, indicating that these relations may not be crucial or could be harmful.
>
> In the BACE task, the impact of removing "tautomer_count" and "covalent_unit_count" was the most pronounced (a decrease of 0.9%), highlighting their importance in this specific prediction task. The minimal impact of removing "indication" (a decrease of 0.2%) suggests that this relation might not be as critical for BACE as for BBBP.
>
> These results provide initial insights into the task-specific importance of different relations in the knowledge graph. We plan to extend this preliminary analysis into a more comprehensive and systematic study in the future, which will allow us to more accurately determine the role and impact of various knowledge graph relations (or entities) on different molecule property prediction tasks.
>
>
>
> **Q2: "Visual grouping of the methods ..."**
>
> Thank you for your suggestion. We have split the non-KG-based and KG-based methods in ***Table 2*** and ***Table 3*** in our revised paper.
>
> **Q3: What affects the steepness of the improvement as knowledge graph size is increased?**
>
> The observations in Figure 3 are derived from a single run, intended to illustrate the impact of KG size on model performance across different tasks. Due to the high computational cost involved — requiring re-training of the K-GNN for each ratio per run and subsequent contrastive learning — the experiment was not scaled to multiple runs. Therefore, the steepness observed in the current Figure 3 may contain an element of randomness, and drawing definitive conclusions from it is challenging. However, we certainly plan to prioritize this aspect as a key area of future research, potentially integrating it with the systematic study outlined in response to Q1.
>
> **Q4: Why wasn’t the Ye et al, 2021 model benchmarked?**
>
> Thanks for this important note! We have benchmarked it (named ***KGE_NFM***) in ***Table 2*** and ***Table 3*** in our revised paper.

---

> > ### Comment · Reviewer_afSU · 2023-11-22
> >
> > I thank the authors for their responses to my questions

---

> > > ### Author Response · Authors · 2023-11-22
> > > **Thank you**
> > >
> > > Thanks a lot for your appreciation of our work! We plan to enhance our paper with the advice you provided, such as including a more extensive version of the experiment for Q1, in our final revision.

---

### Meta-Review · Area_Chair_bqNd · 2023-12-08

**Metareview:**

In this submission, the authors propose a bi-level contrastive learning method for molecular representation, leveraging a high-level knowledge graph to enhance the semantics of molecular representations. Two GNNs are pre-trained to encode the high-level relations within a molecular knowledge graph and the low-level structural information within each molecule, respectively. The two kinds of information are fused and work for downstream tasks, achieving improvements in many learning tasks.

Strengths: (1) The motivation of this work is reasonable, and the paper is well-written and easy to follow. (2) The technical route applied in this work is reasonable, in my opinion.

Weaknesses: Two reviewers have concerns about the implementation of baselines, making the fairness of comparison experiments questionable. Although the authors made efforts to solve the problem, in the discussion phase, AC confirmed with one of the reviewers that the performance of KANO was not successfully reproduced.

**Justification For Why Not Higher Score:**

The superiority of the proposed method is not convincing --- at least two reviewers think that the performance of some baselines is likely to be underestimated, and this concern is not resolved after the rebuttal phase.

**Justification For Why Not Lower Score:**

N/A

---

### Decision · Program_Chairs · 2024-01-16

Reject